_Article_

# Tumor-secreted clusterin promotes cachectic fat wasting via disrupting circadian gene expression and adipogenesis

Yan Liu [1,2]✉, Yehui Zhou[3], Mengmeng Zhang[1,2], Jin Zhang[1,2], Jiahui Chen[1,2], Long Chen[1,2], Jia Tian[1,2], Xiang Lv[1,2], Xinxing Ma[4], Jing Xu[5], Jingwei Shi[6] & Liming Chen [1,7,8]✉

## Abstract

Fat mass loss is a severe complication in cancer-associated cachexia, but its underlying mechanisms remain unclear. This study identifies the tumor-secreted chaperone clusterin (CLU) as a driver of white adipose tissue (WAT) depletion in triple-negative breast cancer (TNBC). CLU secretion is increased in the serum of cachectic TNBC patients. Mechanistically, extracellular clusterin scavenges 14-3-3 in WAT, inhibiting nucleocytoplasmic translocation of the molecular clock activator BMAL1, and perturbing the transcriptional repression of circadian rhythm genes, including _PER3_. In tumors, desmosomal protein plakophilin 3 (PKP3) controls CLU stability by competitively binding to its lysosomal receptor LRP2, increasing CLU distribution in plaques and inhibiting its lysosomal degradation. In advanced TNBC patients, increased amounts of secreted CLU, PKP3 and PER3 are associated with cachectic fat loss. Finally, a targeted reduction of PKP3 or CLU in the serum restores PER3 expression rhythmicity and inhibits cachectic adipose wasting in a TNBC mouse model. Taken together, our results identify a targetable a clinically accessible PKP3-clusterin axis that disrupts circadian gene expression in fat tissue in breast cancer.

**Keywords** Fat Mass Loss; Cachexia; Circadian; CLU; PKP3
**Subject Categories** Cancer; Chromatin, Transcription & Genomics; Musculoskeletal System

## Introduction

Cachexia, a common complication in advanced cancer patients, causes the poor quality of life and death in cancer patients (Argiles et al, 2014; Fearon et al, 2012a). Cancer cachexia is a metabolic disorder characterized by involuntary loss of body weight, which often involves the excessive depletion of adipose tissue, also known as fat tissue (Morigny et al, 2021; Yeom and Yu, 2022). However, there are no effective therapies to prevent or reverse cachectic fat mass loss in cancer patients because the underlying mechanism remains unclear.

There are two major types of adipose tissue: white adipose tissue (WAT) and brown adipose tissue (BAT). WAT and BAT usually perform opposite physiological functions, with WAT responsible for energy storage by storing lipids in intracellular lipid droplets and BAT responsible for energy expenditure by oxidizing lipids to fuel thermogenesis (Cinti, 2005). Obesity researchers have explored ways to convert the body's energy-storing WAT into energy-burning BAT, known as WAT browning. However, in advanced cancer patients, WAT browning precedes cancer cachexia, and targeting tumor production of interleukin-6 (IL-6) or tumor-derived parathyroid-hormone-related protein (PTHrP) or their upregulated uncoupling protein 1 (UCP1) in WAT can reduce WAT browning to ameliorate the severity of cachexia (Das et al, 2011; Fearon et al, 2012b; Kir et al, 2014; Petruzzelli et al, 2014). WAT atrophy due to fat mass loss has been emerged as one of the hallmarks of cachexia. However, the underlying mechanism has not been fully elucidated.

Circadian rhythms are generated endogenously by genetically encoded molecular clocks, whose components cooperate to generate cyclic changes in their own abundance and activity, with a periodicity of about one day (Patke et al, 2020). It's well established that circadian locomotor output cycles kaput (CLOCK) and basic helix-loop-helix ARNT-like protein 1 (BMAL1) form the rhythm conductor CLOCK/BMAL1 complex to activate the rhythmic expression of circadian rhythm-related genes, including _PER_ genes such as _PER3_ (Zhang et al, 2014b). There are two different ways for the CLOCK/BMAL1 complex to rhythmically regulate its target genes: (1) the expression level of CLOCK/BMAL1 is rhythmically controlled; (2) the nuclear-cytoplasmic translocation of CLOCK/BMAL1 is rhythmically controlled. 14-3-3 plays an important role in the function of CLOCK/BMAL1 by binding to

[1]Jiangsu Cancer Hospital, Jiangsu Institute of Cancer Research, Affiliated Cancer Hospital of Nanjing Medical University, Nanjing 210009, P. R. China. [2]Department of Biochemistry, School of Life Sciences, Nanjing Normal University, Nanjing 210023, P. R. China. [3]Department of General Surgery, The First Affiliated Hospital of Soochow University, Suzhou 215000, P. R. China. [4]Department of Radiology, The First Affiliated Hospital of Soochow University, Suzhou 215006, P. R. China. [5]Department of Clinical Laboratory, Zhongda Hospital Southeast University, Nanjing 210009, P.R. China. [6]Department of Thoracic Surgery, Nanjing Drum Tower Hospital, Affiliated Hospital of Medical School, Nanjing University, Nanjing 210000, China. [7]Jiangsu Key Laboratory of Innovative Cancer Diagnosis & Therapeutics, Cancer Institute of Jiangsu Province, Nanjing 210009, P. R. China. [8]The Fourth Clinical College of Nanjing Medical University, Nanjing 210009, P. R. China. ✉E-mail: llliuyan@sina.com; chenliming1981@u.nus.edu

BMAL1 to inhibit or delay the entry of BMAL1 into the nucleus to activate the expression of circadian rhythm-related genes (Dang et al, 2016b). Previous studies have shown that perturbed or misaligned circadian rhythms are associated with the increased risk of disease, including cancer, obesity and other metabolic disorders (Ruan et al, 2021). Adipose tissue activity is circadian in nature and adipose metabolic processes are controlled by circadian rhythms (Bass, 2024). However, the relationship between circadian dysregulation and cachexia remains unknown.

Cachexia is common in patients with advanced cancer, but there are no approved drug therapies and no effective medical intervention (Baracos et al, 2018). Breast cancer is the most common cancer in women worldwide, among which triple-negative breast cancer (TNBC) is the most malignant type of breast cancer with a high rate of cachexia (Argiles et al, 2014; Hu et al, 2023). In this study, using a TNBC model of cancer cachexia, we showed that tumor-secreted clusterin (CLU) plays an important role in wasting through continuously activating the expression of circadian rhythm genes, including *PER3*, to perturb or misalign the circadian rhythm in WAT. Mechanistically, plakophilin 3 (PKP3), a key component of desmosomes in cell–cell contacts (Bonne et al, 2003; Todorovic et al, 2014), binds and stabilizes CLU in tumor desmosomes to produce secreted CLU, which enters the WAT to competitively bind with 14-3-3 to inhibit its interaction with CLOCK/BMAL1, resulting in CLOCK/BMAL1 remaining in the nucleus to continuously activate circadian rhythm genes. Clinically, in advanced TNBC patients with cachexia, the majority (70%) of patients have PKP3 and CLU double-positive (PKP3+/CLU+) tumors, which positively correlates with elevated serum CLU levels and elevated *PER3* expression levels in WAT. Therapeutically, targeting the PKP3-CLU axis by PKP3 or CLU suppression effectively ameliorates cachexia in preclinical TNBC patient-derived xenograft (PDX) mice. Taken together, this study establishes the important role of tumor-secreted CLU and its perturbed circadian rhythm in cachectic fat mass loss of WAT and highlights them as promising biomarkers and therapeutic targets for cancer cachexia.

# Results

## Tumor-secreted CLU promotes fat mass loss in cancer cachexia

To identify driver factors for the development of cancer cachexia, we profiled and compared secreted protein expressions in serum collected from women with TNBC-associated cachexia and healthy women using mass spectrometry (MS). The result reveals several differentially expressed proteins in the serum, including 8 upregulated and 4 downregulated proteins in women with TNBC-associated cachexia versus healthy women using the criteria |log2 (Fold Change)| > 1 and $p < 0.05$ (Fig. 1A and Dataset EV1). Serum CLU, also known as secreted CLU, was identified to be one of the significantly upregulated proteins (Fig. 1A and Dataset EV1). We confirmed that the protein levels of secreted CLU in the serum of advanced TNBC patients with cachectic weight loss were significantly higher than those of TNBC patients without cachexia and healthy women, including underweight healthy women with body mass index (<18.5) and normal-weight healthy women with

normal body mass index (18.5 to 24.9), providing serum CLU as a potential biomarker for TNBC cachexia (Fig. 1B). Our previous study showed that secreted CLU, which was elevated in the serum of TNBC mice and patients, was produced by TNBC cells (Liu et al, 2021). We then sought to ascertain whether and how tumor-secreted CLU contributes to the development of cancer cachexia. To investigate this point, we used the TNBC cancer cell line MDA-MB-468 to generate a TNBC cachectic mouse model. The results show that MDA-MB-468 tumors with CLU knockdown by shCLU lentivirus showed significantly decreased serum CLU and alleviated cachectic weight loss compared to the control (Fig. 1C,D). Cachexia is a multifactorial disease characterized by weight loss via muscle and fat loss (Ferrer et al, 2023). Fat mass loss is both a marker and a cause of cachexia (Argiles et al, 2023). However, its role in cachexia has received less attention compared to muscle loss. When we examined the fat mass in our MDA-MB-468 TNBC cachexia mouse models, we found that MDA-MB-468 tumors with CLU knockdown significantly increased the weight and size of inguinal white adipose tissue (iWAT) and gonadal white adipose tissue (gWAT), other than brown adipose tissue (BAT), compared to controls (Figs. 1E,F and EV1). Mice bearing CLU knockdown tumors have larger white adipocytes full of stored fats compared to the control (Fig. 1G). When we examined the level of tumor-secreted CLU in iWAT, the results showed that knockdown of tumor CLU led to a decrease in the level of secreted CLU in iWAT compared to the control (Fig. 1H). These results suggest that tumor induces fat mass loss through tumor-secreted CLU in WAT.

## Secreted CLU promotes cachectic fat mass loss via disrupting circadian rhythm in WAT

To explore the mechanism underlying tumor-secreted CLU in promoting the cachectic fat mass loss of WAT, we examined the alteration of transcriptomes in the iWAT of mice bearing MDA-MB-468 tumors with or without CLU knockdown by RNA-seq (Dataset EV2). Interestingly, a gene ontology-biological process analysis on the RNA-seq data revealed that 6 of the top 10 enriched biological processes were related to the circadian rhythm, and 8 known circadian rhythm genes were found to be differentially expressed between the iWATs derived from mice bearing MDA-MB-468 TNBC tumor with or without CLU knockdown (Fig. 2A,B and Dataset EV2 and EV3). These 8 genes were downregulated genes, including Per genes such as Per2 and Per3. Consistent with the RNA-seq data, RT-qPCR results showed that these 8 genes were downregulated in the iWAT of mice bearing CLU-knockdown MDA-MB-468 tumors compared to the control (Fig. 2B,C). *Per* genes, including *Per3*, are rhythmically transcribed by the rhythm conductor complex, Clock/Bmal1 (Ruan et al, 2021). The RNA-seq data and RT-qPCR results consistently showed that neither *Clock* nor *Bmal1* were downregulated in the iWAT of mice CLU-knockdown MDA-MB-468 tumors compared to the control (Fig. 2C). We then investigated whether and how tumor-secreted CLU affects the rhythmic expression of *Clock*, *Bmal1* and *Per3* in the iWAT. To address this, we examined the expression of *Clock*, *Bmal1* and *Per3* in the iWAT of mice bearing MDA-MB-468 TNBC tumors with or without CLU knockdown at different time points in a 48-hour cycle. The results showed that *Clock* and *Bmal1* exhibited normal rhythmic expression in the iWAT of mice bearing MDA-MB-468 tumors with and without CLU knockdown (Fig. 2D). In

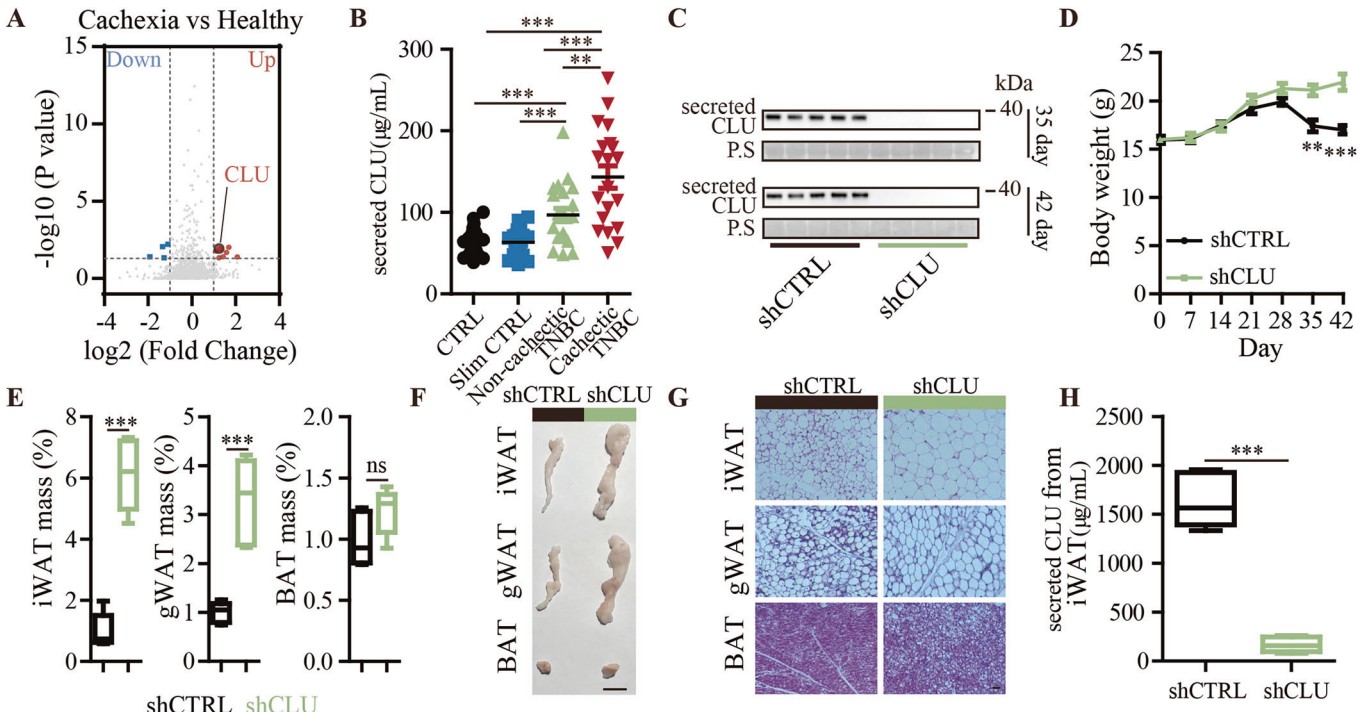

**Figure 1. Identification of tumor-secreted CLU as a causal factor of cachectic fat mass loss in TNBC models.**

(A) A volcano plot represents the expression of all proteins: significantly downregulated and upregulated proteins (|log2 FC| >1 and $P < 0.05$) are colored blue and red using unpaired two-tailed Student's t tests, respectively; proteins that do not show significant expression changes are colored gray (Healthy, $n = 22$; Cachexia, $n = 20$). (B) Secreted CLU levels in serum samples from cachectic TNBC patients compared to healthy women, slim healthy women, and non-cachectic TNBC patients by ELISA assay. (CTRL, $n = 22$; Slim CTRL, $n = 21$; Non-cachectic TNBC, $n = 21$; Cachectic TNBC, $n = 20$; CTRL vs Non-cachectic TNBC, $P = 0.0009$; CTRL vs Cachectic TNBC, $P < 0.0001$; Slim CTRL vs Non-cachectic TNBC, $P = 0.0007$; Slim CTRL vs Cachectic TNBC, $P < 0.0001$; Non-cachectic TNBC vs Cachectic TNBC, $P = 0.0055$). (C–H) MDA-MB-468 tumor-bearing mouse model with and without CLU knockdown ($n = 5$ per group): (C) Representative Western blotting images showing serum CLU levels at 35 days, 42 days after inoculation of cancer cells into mouse models; (D) Body weight of mice over the time after cancer cell inoculation (35 day, $P = 0.0023$; 42 day, $P = 0.0008$); (E) Change of iWAT, gWAT, BAT weight (shCTRL vs shCLU: iWAT, $P < 0.0001$; gWAT, $P = 0.0005$; BAT, $P = 0.1135$; $n = 5$ per group), in a box-plot, the center line represents the median of the data, while the lower and upper limits of the box correspond to the first quartile and third quartile, respectively; (F) Representative iWAT, gWAT, BAT photos, Scale bar: 1 cm; (G) Representative H&E staining images of iWAT, gWAT, BAT, Scale bar: 50 µM; (H) Secreted CLU levels in iWAT by ELISA assay (shCTRL vs shCLU, $P < 0.0001$; $n = 5$ per group), in a box-plot, the center line represents the median of the data, while the lower and upper limits of the box correspond to the first quartile and third quartile, respectively. The data are presented as mean ± SEM ($n = $ [X] biologically independent samples). $**p < 0.01$, $***p < 0.001$, ns for not significant. Statistical comparisons between groups were performed using unpaired two-tailed Student's t tests. Source data are available online for this figure.

comparison, rhythmic expression of *Per3* was found only in the iWAT of mice bearing MDA-MB-468 tumors with CLU knockdown, whereas *Per3* showed a loss of rhythm and aberrant overexpression in the iWAT of mice bearing MDA-MB-468 tumors without CLU knockdown compared to those with CLU knockdown (Fig. 2D). Consistently, Western blot showed that tumor CLU knockdown can promote the recovery of the circadian rhythmic expression of Per3 in iWAT (Fig. 2E). It has been reported that 14-3-3 regulates the rhythmic expression of Clock/Bmal1 downstream genes, including *Per3*, by binding to Clock/Bmal1 to inhibit the entry of this rhythm conductor complex into the nucleus to activate gene expression (Dang et al, 2016a; Luciano et al, 2018). When we carried out Co-IP by using 14-3-3 antibodies in the iWAT of mice bearing MDA-MB-468 TNBC tumor with or without CLU knockdown. The results indicate that protein level of Clock/Bmal1 pulled-down by the 14-3-3 antibody is significantly increased in iWAT of mice bearing CLU knockdown tumors compared to controls (Figs. 2F and EV2A). Furthermore, nucleocytoplasmic separation assays show that the protein level of Clock/Bmal1 increased in the nucleus in iWAT cells with 14-3-3 knockdown

compared to iWAT without 14-3-3 knockdown, suggesting that 14-3-3 plays an important role in preventing Bmal1/Clock protein entry to the nucleus (Fig. EV2B). Moreover, 14-3-3 knockdown restored the transcription level of Per3 in iWAT in mice bearing CLU knockdown tumors compared to control (Fig. EV2C). These results suggest that tumor-secreted CLU promotes *Per3* to show loss of rhythm and aberrant overexpression in the iWAT by competitively binding to 14-3-3 to inhibit its interaction with Clock/Bmal1, leading to the dysregulated entry of Clock/Bmal1 into the nucleus for perpetually activating the expression of circadian rhythm-related genes, including *Per3*. To further explore whether aberrant overexpression of *Per3* by secreted CLU underlies fat mass loss, we examined the iWAT of the healthy mice with addition secreted CLU treatment by intravenous injection of secreted CLU purified protein with and without additional knockdown *Per3* in the iWAT. The results showed that intravenous injection of secreted CLU purified protein can enter iWAT and cause *Per3* to show loss of rhythm and aberrant overexpression in the iWAT (Figs. 2G–I and EV2D). For fat mass loss, the intravenous injection of secreted CLU purified protein significantly decreased both the

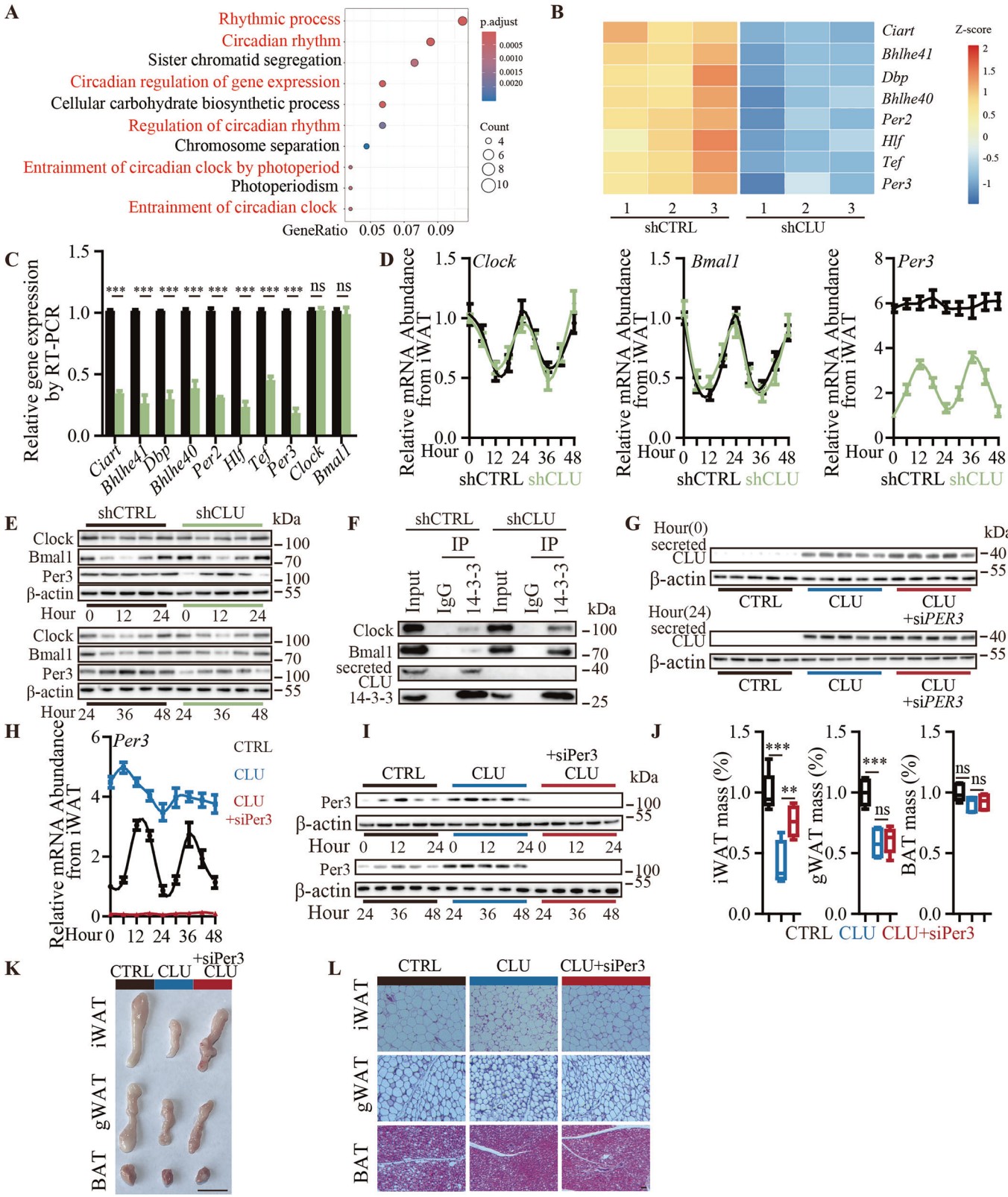

**Figure 2.   Disruption of circadian rhythm by tumor-secreted CLU to induce cachectic fat mass loss.**

(A–G) iWAT of MDA-MB-468 tumor-bearing mice with compared to without CLU knockdown ($n = 5$ per group): (A) GO enrichment analysis of the differentially expressed genes with a hypergeometric test; (B) Heatmap showing the differential expression of circadian rhythm-related genes as indicated; (C) Circadian rhythm-related gene expression by RT-qPCR (shCTRL vs shCLU: *Ciart*, $P < 0.0001$; *Bhlhe41*, $P = 0.0001$; *Dbp*, $P = 0.0001$; *Bhlhe40*, $P = 0.0003$; *Per2*, $P < 0.0001$; *Hlf*, $P < 0.0001$; *Tef*, $P < 0.0001$; *Per3*, $P < 0.0001$; *Clock*, $P = 0.9363$; *Bmal1*, $P = 0.7469$). (D, E) CLOCK, BMAL1, PER3 expression in a 48-h cycle by (D) RT-qPCR and (E) Western blot assay; (F) Representative Western blot images of CLOCK, BMAL1, secreted CLU and 14-3-3 by Co-IP assays. (G–L) iWAT of healthy nude mice with and without intravenous injection of purified secreted CLU and WAT injection of siPER3 ($n = 5$ per group): (G) Representative Western blot images showing tumor-secreted CLU levels; (H, I) PER3 expression in a 48-hour cycle by (H) RT-qPCR and (I) Western blot assay; (J) Weight change (CTRL vs CLU: iWAT, $P = 0.0005$; gWAT, $P = 0.0003$; BAT, $P = 0.0527$; CLU vs CLU+siPer3: iWAT, $P = 0.0063$; gWAT, $P = 0.7527$; BAT, $P = 0.7305$), in a box-plot, the center line represents the median of the data, while the lower and upper limits of the box correspond to the first quartile and third quartile, respectively; (K) Representative iWAT, gWAT, BAT photos, Scale bar: 1 cm; (L) Representative H&E staining images iWAT, gWAT, BAT, Scale bar: 50 μM. The data are presented as mean ± SEM ($n = $ [X] biologically independent samples). *$p < 0.05$, **$p < 0.01$, ***$p < 0.001$, ns for not significant. Statistical comparisons between groups were performed using unpaired two-tailed Student's t tests. Source data are available online for this figure.

weight and size of WAT compared to the control, and additional knockdown of *Per3* in iWAT alleviated the secreted CLU-promoted fat mass loss (Figs. 2J–L and EV2E). It has been reported that the deletion of *Per3* gene promotes adipogenesis in vivo (Aggarwal et al, 2017). We hypothesized that the dysregulation of Per3-regulated adipogenesis contributes to the CLU-promoted fat mass loss. To test this hypothesis, we first examined the expression of adipogenesis-related genes by RT-qPCR. The results showed increased expression levels of selected adipogenesis-related genes were increased in the iWAT of mice with CLU-knockdown tumors compared to controls (Fig. EV3A). In comparison, intravenous injection of secreted CLU purified protein significantly decreased adipogenesis gene expression compared to the control, where additional knockdown of *Per3* in iWAT restored adipogenesis gene expression levels (Fig. EV3B). Oil Red O staining assays showed consistent results (Fig. EV3C,D). In addition to adipogenesis dysregulation, fat mass loss can be promoted by other mechanisms, such as increased energy expenditure (Cristancho and Lazar, 2011). Using a metabolic cage assay, we explored energy expenditure and found decreased oxygen consumption, carbon dioxide production, and heat generation in mice with CLU-knockdown tumors compared to controls (Fig. EV3E–G). Intravenous injection of secreted CLU purified protein significantly increased the oxygen consumption, carbon dioxide production, and heat generation of mice compared to the control, and additional knockdown of *Per3* in iWAT alleviated the energy expenditure (Fig. EV3H–J). These results suggest that both adipogenesis and energy expenditure contribute to fat mass loss promoted by CLU-Per3 axis. Further research is needed to explore if other factors contribute to circadian rhythm-regulated fat mass loss. Taken together, these results strongly suggest that the disruption of the circadian rhythms and the rhythmic expression of *Per3* in WAT by tumor-secreted CLU promotes cachectic fat mass loss.

## PKP3 stabilizes CLU by promoting desmosome-inhibited lysosomal degradation

Next, we intended to identify the upstream regulator of tumor-secreted CLU. Since secreted CLU is the mature form of CLU in cells, to address this point, we first performed co-immunoprecipitation (Co-IP) combined with mass spectrometry to reveal the CLU interactome in MDA-MB-468 TNBC cells. PKP3, a key desmosomal protein in cell–cell contacts, was identified as a CLU-binding protein (Dataset EV4). Co-IP assays followed by Western blotting validated the physical interaction between endogenous CLU and PKP3 in cells,

including CLU and PKP3 in TNBC cells (Figs. 3A and EV4A). Domain analysis revealed that CLU includes a signal peptide followed by α and β chain domains, secreted CLU contains α and β chain domains, and PKP3 contains arm and head chain domains (Fig. EV4B). We further overexpressed HA-tagged CLU or its series of truncated mutant fragments in HEK293T cells with ectopic overexpression of FLAG-tagged PKP3, performed Co-IP using anti-FLAG and anti-HA antibodies. The results showed that the CLU α and β chain domains and the PKP3 arm and head chain domains all contribute to the physical interaction between CLU and PKP3 (Fig. EV4C,D). To explore the regulatory relationship between CLU and PKP3, we examined the expression of CLU and the expression of PKP3 upon knockdown of PKP3 and CLU in both investigated TNBC cells. The results showed that PKP3 knockdown decreased the CLU protein but not mRNA level, while CLU knockdown didn't significantly affect PKP3 expression level (Figs. 3B,C and EV4E–J). These results suggest that PKP3 is an upstream of CLU to unidirectionally upregulate CLU protein level in TNBC cells. We then hypothesized that PKP3 upregulates CLU by stabilizing CLU. To address this, we performed a protein stability assay by treating TNBC cells with cycloheximide (CHX). The results confirmed that PKP3 knockdown decreased the stability of CLU in TNBC cells (Figs. 3D and EV4K). We then investigated how PKP3 stabilizes CLU, through the ubiquitin-proteasome system (UPS) or the autophagy-lysosome pathway (ALP) (Amaravadi et al, 2016; Manasanch and Orlowski, 2017). To address this issue, we used the UPS inhibitor MG132 and the ALP inhibitor chloroquine (CQ). The results showed that CQ, but not MG132, reversed the CLU degradation upon PKP3 knockdown (Figs. 3E and EV4L). This result suggests that PKP3 stabilizes CLU by inhibiting lysosomal degradation. PKP3 is a key structural component in the cytoplasmic portion of desmosomes, which are the major intercellular junctions in vertebrate intercellular contacts (Abercrombie, 1967; Bonne et al, 2003; Delva et al, 2009; Eagle and Levine, 1967; Todorovic et al, 2014). Desmoglein (DSG) is a well-documented extracellular marker of desmosomes in intercellular contacts (Sikora et al, 2020). We found that PKP3 was required for the formation of desmosomes in intercellular contacts of TNBC cells, and CLU colocalized with PKP3 at the desmosome-like region (Figs. 3F,G and EV4M–R). These results led us to hypothesize that PKP3 inhibits lysosomal degradation of CLU by increasing and decreasing the desmosomal and lysosomal distribution of CLU, respectively. To address this hypothesis, we investigated whether and how PKP3 knockdown affects the lysosomal distribution of CLU by examining the alteration of CLU colocalizing with LAMP1, a well-documented lysosomal marker (Wang et al, 2013). The results showed

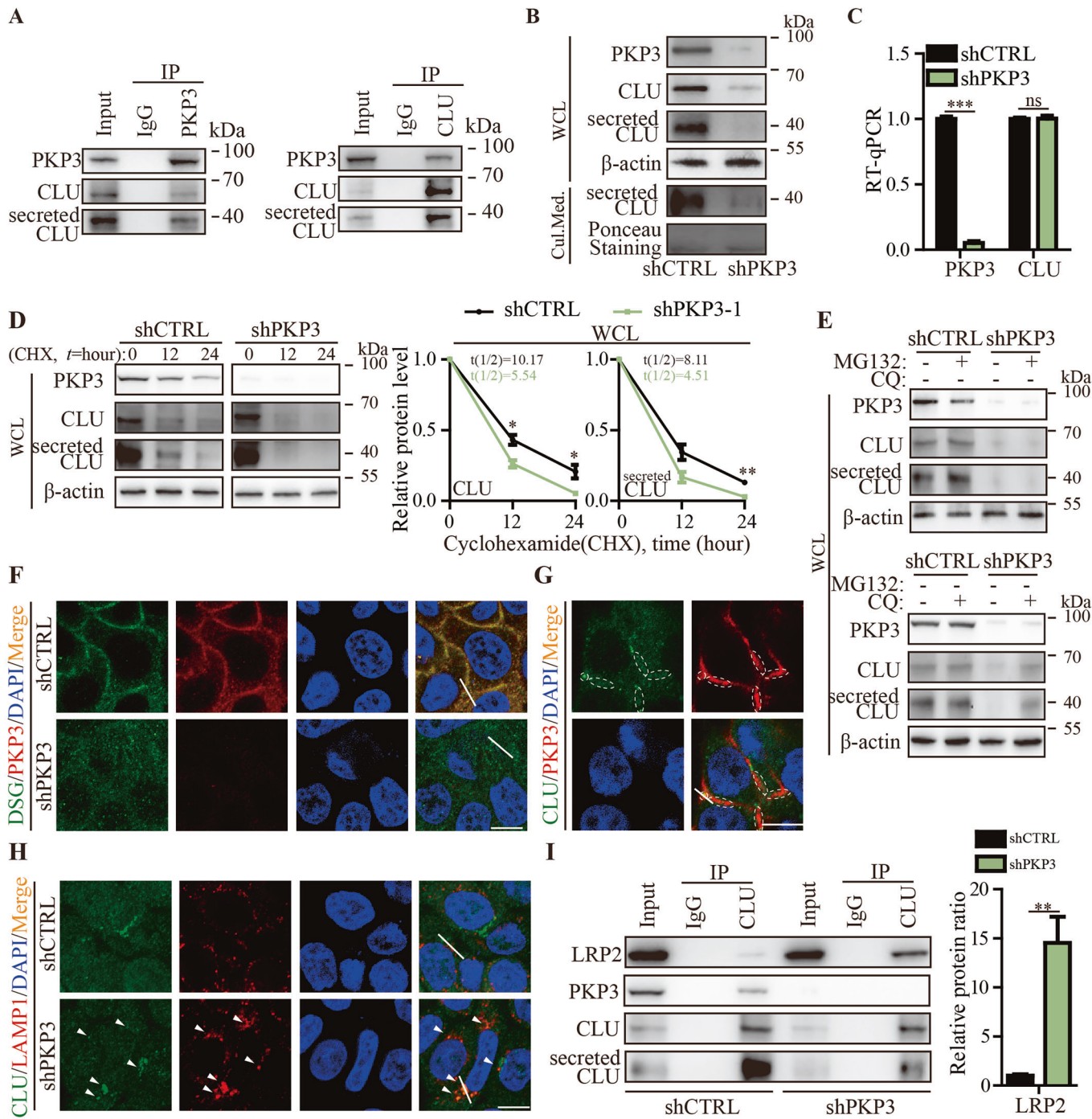

that PKP3 knockdown increased the lysosomal distribution of CLU compared to controls (Figs. 3H and EV4S–T). LRP2 is a lysosomal receptor that is responsible for recruiting CLU to lysosomes for lysosomal degradation (Kim et al, 2011; Liu et al, 2021). We found that the LRP2 protein levels, which was pulled-down by CLU antibody, was elevated in PKP3-knockdown TNBC cells compared to controls (Figs. 3I and EV4U). Taken together, these results suggest that the formation of intercellular desmosomes in a PKP3-coordinated manner distributes PKP3 to desmosomal plaques in cell–cell contact, where PKP3 captures CLU at the desmosomal region to reduce its lysosomal distribution and degradation.

## PKP3 promotes cachectic fat mass loss via tumor-secreted CLU

We then investigated whether PKP3, as an upstream of CLU in TNBC cells, promotes cachectic fat mass loss via tumor-secreted CLU to disrupt the circadian rhythm in WAT. To address this point, we generated TNBC cachexia mouse models using two TNBC cell lines, with PKP3 knockdown, or CLU knockdown, or PKP3 knockdown with ectopically overexpressed CLU compared to control. The results showed that mice bearing PKP3-knockdown and CLU-knockdown TNBC tumors showed significant amelioration of weight loss

**Figure 3.   CLU stabilization by PKP3 in TNBC cancer cells.**

(A) Representative Western blot images showing PKP3, CLU and secreted CLU levels in Co-IP assays using TNBC MDA-MB-468 cells. (B) Representative Western blot images showing PKP3, CLU and secreted CLU protein levels in MDA-MB-468 cells and secreted CLU protein levels in cell culture medium (Cul. Med.) upon PKP3 knockdown compared to controls. (C) PKP3 and CLU mRNA levels in MDA-MB-468 cells by RT-qPCR (shCTRL vs shPKP3: *PKP3*, $P < 0.0001$; *CLU*, $P = 0.9436$; $n = 4$ per group). (D) Representative Western blot images and their quantification of PKP3, CLU and secreted CLU protein levels in MDA-MB-468 cells upon PKP3 knockdown compared to control after CHX treatment. (shCTRL vs shPKP3: CLU, 12-hour, $P = 0.0174$; 24-hour, $P = 0.0360$; secreted CLU, 24-hour, $P = 0.0047$; $n = 3$ per group). (E) Representative Western blot images showing PKP3, CLU and secreted CLU protein levels in cancer cells upon MG132 or CQ treatment in PKP3 knockdown MDA-MB-468 cells compared to control. (F) Representative immunofluorescence images showing PKP3 and DSG localization in MDA-MB-468 cells after PKP3 knockdown compared to the control. Green for DSG, red for PKP3, blue for DAPI and orange for DSG-PKP3 colocalization. Scale bar: 10 μM. (G) Representative immunofluorescence images showing PKP3 and CLU localization in MDA-MB-468 cells. White dashed lines highlight the intercellular plaque regions, green for CLU, red for PKP3, blue for DAPI, orange for CLU-PKP3 colocalization. Scale bar: 10 μM. (H) Representative immunofluorescence images showing CLU and LAMP1 localization in MDA-MB-468 cells upon PKP3 knockdown compared to the control. Green for CLU, red for LAMP1, blue for DAPI and orange for CLU-LAMP1 colocalization. Arrows indicated CLU-LAMP1 colocalization foci. Scale bar: 10 μM. (I) Representative Western blot images and quantification of LRP2, PKP3, CLU and secreted CLU levels in Co-IP assays using MDA-MB-468 cells upon PKP3 knockdown compared to the control (shCTRL vs shPKP3: *LRP2*, $P = 0.0070$; $n = 3$ per group). The data are presented as mean ± SEM ($n = $ [X] biologically independent samples). $*p < 0.05$, $**p < 0.01$, $***p < 0.001$, ns for not significant. Statistical comparisons between groups were performed using unpaired two-tailed Student's t tests. Source data are available online for this figure.

compared to control, while PKP3-knockdown with additional ectopic overexpression of CLU induced weight loss compared to PKP3-knockdown without additional ectopic overexpression of CLU (Figs. 4A and EV5A). When we examined the fat mass, we consistently found that PKP3 knockdown and CLU knockdown significantly increased the weight and size of WAT compared to control, whereas PKP3 knockdown with additional ectopic overexpression of CLU induced cachectic fat mass loss compared to PKP3 knockdown without additional ectopic overexpression of CLU (Figs. 4B,C and EV5B,C). PKP3 knockdown and CLU knockdown have larger white fat cells full of stored fats in WAT compared to control and PKP3-knockdown with additional ectopic overexpression of CLU (Figs. 4D and EV5D). When we examined the level of tumor-secreted CLU in iWAT, the results showed that tumor CLU knockdown and tumor PKP3 knockdown decreased tumor-secreted CLU in iWAT compared to control, and ectopic overexpression of CLU in tumor cells with PKP3 knockdown can restore tumor-secreted CLU in iWAT (Figs. 4E and EV5E). These results suggest that PKP3 induces fat mass loss by acting as an upstream stabilizer of CLU in TNBC cells to produce tumor-secreted CLU in WAT. We then further investigated whether and how PKP3 disrupts the circadian rhythm in WAT for the development of cachectic fat mass loss by tumor-secreted CLU. To address this point, we first used RT-qPCR to examine the expression of the 8 circadian rhythm-related genes, including *Per3*, that were aberrant overexpressed in the WAT of cachexia mice bearing MDA-MB-468 tumors without CLU knockdown. The results showed that in both MDA-MB-468 cachexia TNBC mouse models, PKP3 knockdown and CLU knockdown downregulated the expression of these 8 genes in the iWAT compared to control, while PKP3-knockdown with additional ectopic overexpression of CLU restored the overexpression of these genes in iWAT compared to PKP3-knockdown without additional ectopic overexpression of CLU, whereas neither *Clock* nor *Bmal1* were downregulated by CLU knockdown or PKP3 knockdown with and without ectopic over-expression of CLU in TNBC tumor cells compared to control (Fig. 4F). We then examined whether and how the rhythmic expression of *Per3*, *Clock*, and *Bmal1* in the WAT was affected by measuring the expression of *Per3*, *Clock*, and *Bmal1* in the iWAT of mice at different time points in a 48-hour cycle. The results showed that, like mice bearing TNBC tumors with and without CLU knockdown, *Clock* and *Bmal1* exhibited normal rhythmic expression in the iWAT of mice bearing TNBC tumors with and without PKP3 knockdown, as well as

PKP3 knockdown with additional overexpression of CLU (Figs. 4G,H and EV5F,G). In comparison, rhythmic expression of *Per3* was found in the iWAT derived from mice bearing TNBC tumors with CLU knockdown or PKP3 knockdown, and overexpression CLU in TNBC tumor with PKP3 knockdown restored *Per3* to show loss of rhythm and aberrant overexpression in the iWAT of mice (Figs. 4I and EV5H). Consistently, Western blot showed that overexpression of CLU can disrupt the circadian rhythmic expression of Per3 in the iWAT of mice bearing TNBC tumors with PKP3 knockdown (Figs. 4J and EV5I). Taken together, these results confirm the driver role of tumor-secreted CLU in cachectic fat mass loss, where CLU stabilization by PKP3 in tumor cells produces tumor-secreted CLU in WAT to disrupt the circadian rhythm and promote fat mass loss.

## Tumor PKP3-CLU axis is associated with cachectic fat mass loss in human TNBC patients

Next, we further investigated the importance and reality of the role of the PKP3-CLU axis in disrupting circadian rhythm and inducing cachectic fat mass loss in human TNBC patients. For this purpose, we collected samples, including primary tumors, sera, and WATs from TNBC patients with cachexia using the criteria of body mass less than<18.5. When we examined the protein levels of PKP3 and CLU in TNBC tumor samples by immunohistochemical analysis, we found that 87% of CLU + TNBC tumors were PKP3+ tumors, while PKP3 and CLU double-negative (PKP3-/CLU-) TNBC accounted for only 15% of TNBC tumors (Fig. 5A,B). When PKP3 and CLU protein levels in TNBC tumor samples were quantified and analyzed, the results showed that PKP3 and CLU protein levels were significantly higher in TNBC tumor samples compared to paracancerous samples and were positively correlated (Fig. 5C–E). Furthermore, when we measured secreted CLU in sera, patients with CLU + TNBC tumors had higher levels of serum secreted CLU than those with CLU- TNBC tumors (Fig. 5F). These results suggest that secreted CLU was derived from tumor CLU. In support, the level of serum secreted CLU was significantly positively correlated with the level of both PKP3 and CLU in the tumors (Fig. 5G,H). These results further support that PKP3 is an upstream of CLU in the tumor for the production of tumor-secreted CLU in the serum. We then investigated the clinical relevance of the circadian rhythm disrupted by tumor-secreted CLU in the induction of cachectic fat mass loss in TNBC patients.

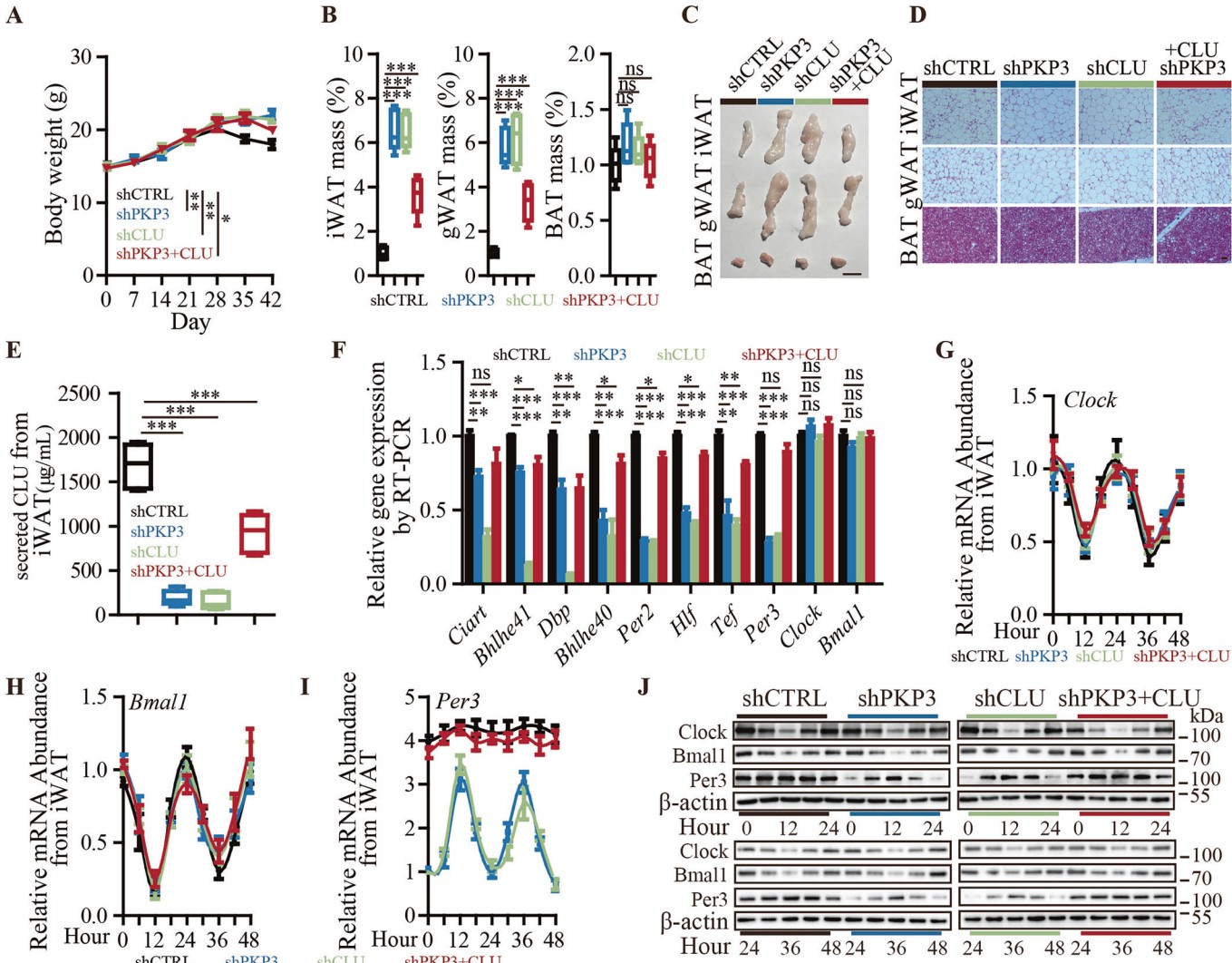

**Figure 4. Disruption of WAT's circadian rhythm by tumor PKP3-CLU axis for cachectic fat mass loss.**

(A–J) MDA-MB-468 tumor-bearing mice with PKP3 knockdown, CLU knockdown, or PKP3 knockdown combined with ectopic overexpression of CLU: (A) Body weight (shCTRL vs shPKP3, $P = 0.0029$; shCTRL vs shCLU, $P = 0.0016$; shCTRL vs shPKP3+CLU, $P = 0.0412$; $n = 5$ per group); (B) Weight change of iWAT, gWAT, BAT (iWAT: shCTRL vs shPKP3, $P < 0.0001$; shCTRL vs shCLU, $P < 0.0001$; shCTRL vs shPKP3+CLU, $P = 0.0002$; gWAT: shCTRL vs shPKP3, $P < 0.0001$; shCTRL vs shCLU, $P < 0.0001$; shCTRL vs shPKP3+CLU, $P = 0.0003$; BAT: shCTRL vs shPKP3, $P = 0.1541$; shCTRL vs shCLU, $P = 0.2282$; shCTRL vs shPKP3+CLU, $P = 0.6989$; $n = 5$ per group), in a box-plot, the center line represents the median of the data, while the lower and upper limits of the box correspond to the first quartile and third quartile, respectively; (C) Representative iWAT, gWAT, BAT photos, Scale bar: 1 cm; (D) Representative H&E staining images of iWAT, gWAT, BAT, Scale bar: 50 μM; (E) Tumor-secreted CLU levels in iWAT by ELISA assay (shCTRL vs shPKP3, $P < 0.0001$; shCTRL vs shCLU, $P < 0.0001$; shCTRL vs shPKP3+CLU, $P = 0.0009$; $n = 5$ per group), in a box-plot, the center line represents the median of the data, while the lower and upper limits of the box correspond to the first quartile and third quartile, respectively; (F) Expression of circadian rhythm-related genes in iWAT by RT-qPCR (*Ciart*: shCTRL vs shPKP3, $P = 0.0037$; shCTRL vs shCLU, $P < 0.0001$; shCTRL vs shPKP3+CLU, $P = 0.1400$; *Bhlhe41*: shCTRL vs shPKP3, $P = 0.0007$; shCTRL vs shCLU, $P < 0.0001$; shCTRL vs shPKP3+CLU, $P = 0.0133$; *Dbp*: shCTRL vs shPKP3, $P = 0.0025$; shCTRL vs shCLU, $P < 0.0001$; shCTRL vs shPKP3+CLU, $P = 0.0085$; *Bhlhe40*: shCTRL vs shPKP3, $P = 0.0003$; shCTRL vs shCLU, $P = 0.0011$; shCTRL vs shPKP3+CLU, $P = 0.0257$; *Per2*: shCTRL vs shPKP3, $P < 0.0001$; shCTRL vs shCLU, $P < 0.0001$; shCTRL vs shPKP3+CLU, $P = 0.0136$; *Hlf*: shCTRL vs shPKP3, $P < 0.0001$; shCTRL vs shCLU, $P < 0.0001$; shCTRL vs shPKP3+CLU, $P = 0.0164$; *Tef*: shCTRL vs shPKP3, $P = 0.0029$; shCTRL vs shCLU, $P < 0.0001$; shCTRL vs shPKP3+CLU, $P = 0.0058$; *Per3*: shCTRL vs shPKP3, $P < 0.0001$; shCTRL vs shCLU, $P < 0.0001$; shCTRL vs shPKP3+CLU, $P = 0.1113$; *Clock*: shCTRL vs shPKP3, $P = 0.2642$; shCTRL vs shCLU, $P = 0.4689$; shCTRL vs shPKP3+CLU, $P = 0.2502$; *Bmal1*: shCTRL vs shPKP3, $P = 0.1956$; shCTRL vs shCLU, $P = 0.7984$; shCTRL vs shPKP3+CLU, $P = 0.7935$; $n = 4$ per group); (G–J) CLOCK, BMAL1, PER3 expression in iWAT in a 48-hour cycle by (G–I) RT-qPCR and (J) Western blot assay ($n = 4$ per group). The data are presented as mean ± SEM ($n = $ [X] biologically independent samples). *$p < 0.05$, **$p < 0.01$, ***$p < 0.001$, ns for not significant. Statistical comparisons between groups were performed using unpaired two-tailed Student's t tests. Source data are available online for this figure.

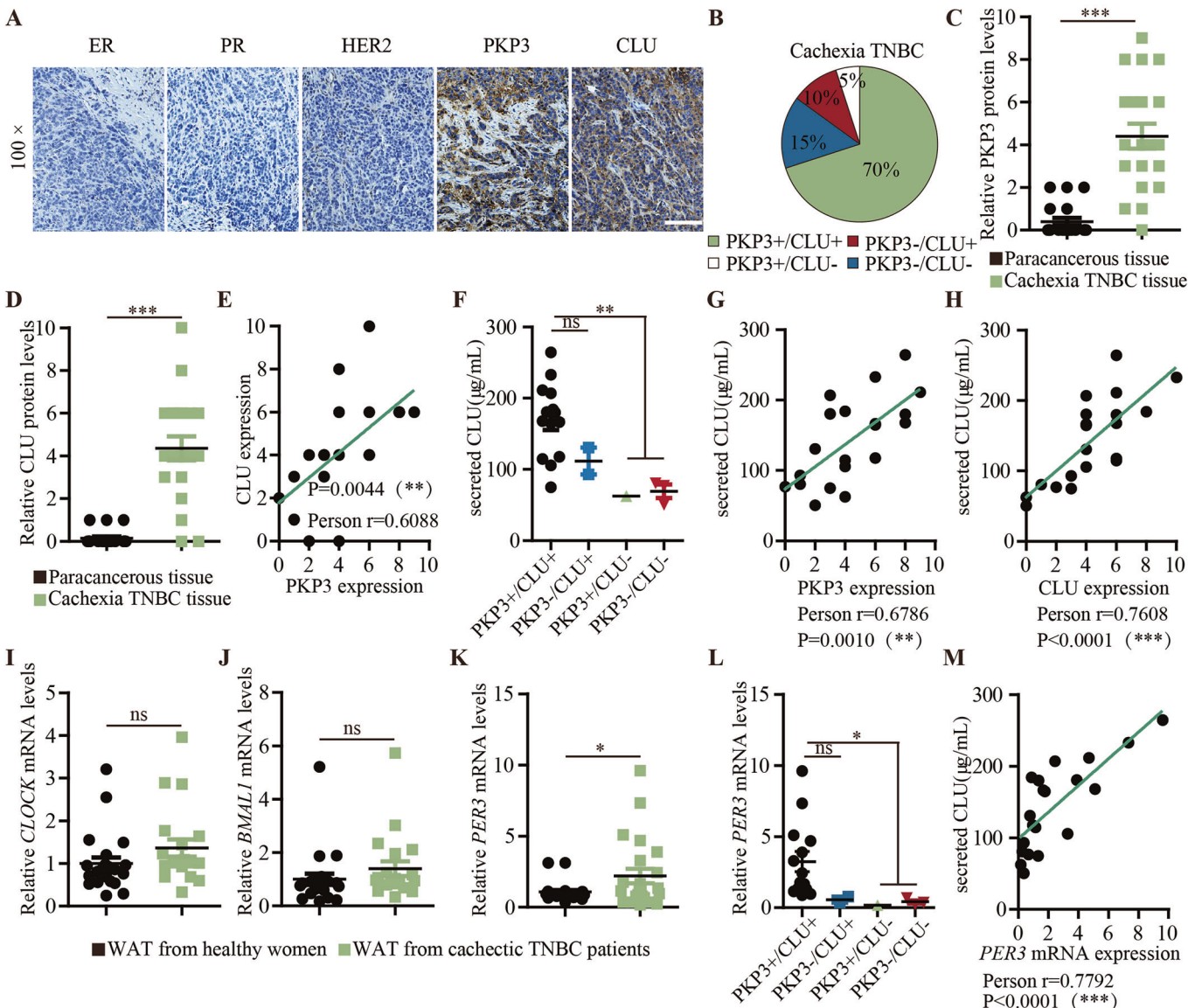

**Figure 5. Clinical relevance.**

(A–E) Tumor samples from TNBC patients with cachexia: (A) Representative images of immunohistochemistry assays showing the expression of ER, PR, HER2, PKP3, CLU in tumors, Scale bar: 100 μM; (B) A pie chart showing the percentage of PKP3 positive (PKP3+), PKP3 negative (PKP3-), CLU positive (CLU+), and CLU negative (CLU-) tumor samples; (C) Quantification of relative PKP3 protein levels in tumor tissues compared to precancerous tissues (Paracancerous tissue vs Cachexia TNBC tissue: *PKP3*, $P < 0.0001$; $n = 20$ per group); (D) Quantification of relative CLU protein levels in tumor tissues compared to precancerous tissues (Paracancerous tissue vs Cachexia TNBC tissue: *CLU*, $P < 0.0001$; $n = 20$ per group); (E) Correlation analysis between PKP3 and CLU protein levels in tumors; (F) Tumor-secreted CLU levels in serum samples from TNBC patients with cachexia (PKP3 +/CLU+ vs PKP3-/CLU +: secreted CLU, $P = 0.1556$; PKP3 +/CLU+ vs PKP3 +/CLU- & PKP-/CLU-: secreted CLU, $P = 0.0016$; $n = 14$ in PKP3 +/CLU+ group; $n = 2$ in PKP3-/CLU+ group; $n = 1$ in PKP3 +/CLU- group; $n = 3$ in PKP3-/CLU- group); (G) Correlation analysis between tumor PKP3 and serum tumor-secreted CLU levels; (H) Correlation analysis between tumor CLU and serum tumor-secreted CLU levels; (I-K) Expression of circadian rhythm-related genes in WAT of cachexia TNBC patients compared to healthy women by RT-qPCR : (I) *CLOCK* (WAT from healthy women vs WAT from cachectic TNBC patients: $P = 0.1386$; $n = 23$ in WAT from healthy women group; $n = 20$ in WAT from cachectic TNBC patients group); (J) *BMAL1* (WAT from healthy women vs WAT from cachectic TNBC patients: $P = 0.2524$; $n = 23$ in WAT from healthy women group; $n = 20$ in WAT from cachectic TNBC patients group); (K) *PER3* (WAT from healthy women vs WAT from cachectic TNBC patients: $P = 0.0139$; $n = 23$ in WAT from healthy women group; $n = 20$ in WAT from cachectic TNBC patients group). (L) *PER3* mRNA level in WAT of cachexia TNBC patients by RT-qPCR (PKP3 +/CLU+ vs PKP3-/CLU +: PKP3, $P = 0.1865$; PKP3 +/CLU+ vs PKP3 +/CLU- & PKP-/CLU-: PKP3, $P = 0.0491$; $n = 23$ in WAT from healthy women group; $n = 20$ in WAT from cachectic TNBC patients group). (M) Correlation analysis between WAT's *PER3* and serum tumor-secreted CLU levels in cachexia TNBC patients. The data are presented as mean ± SEM. *$p < 0.05$, **$p < 0.01$, ***$p < 0.001$, ns for not significant. Statistical comparisons between groups were performed using unpaired two-tailed Student's t tests. Source data are available online for this figure.

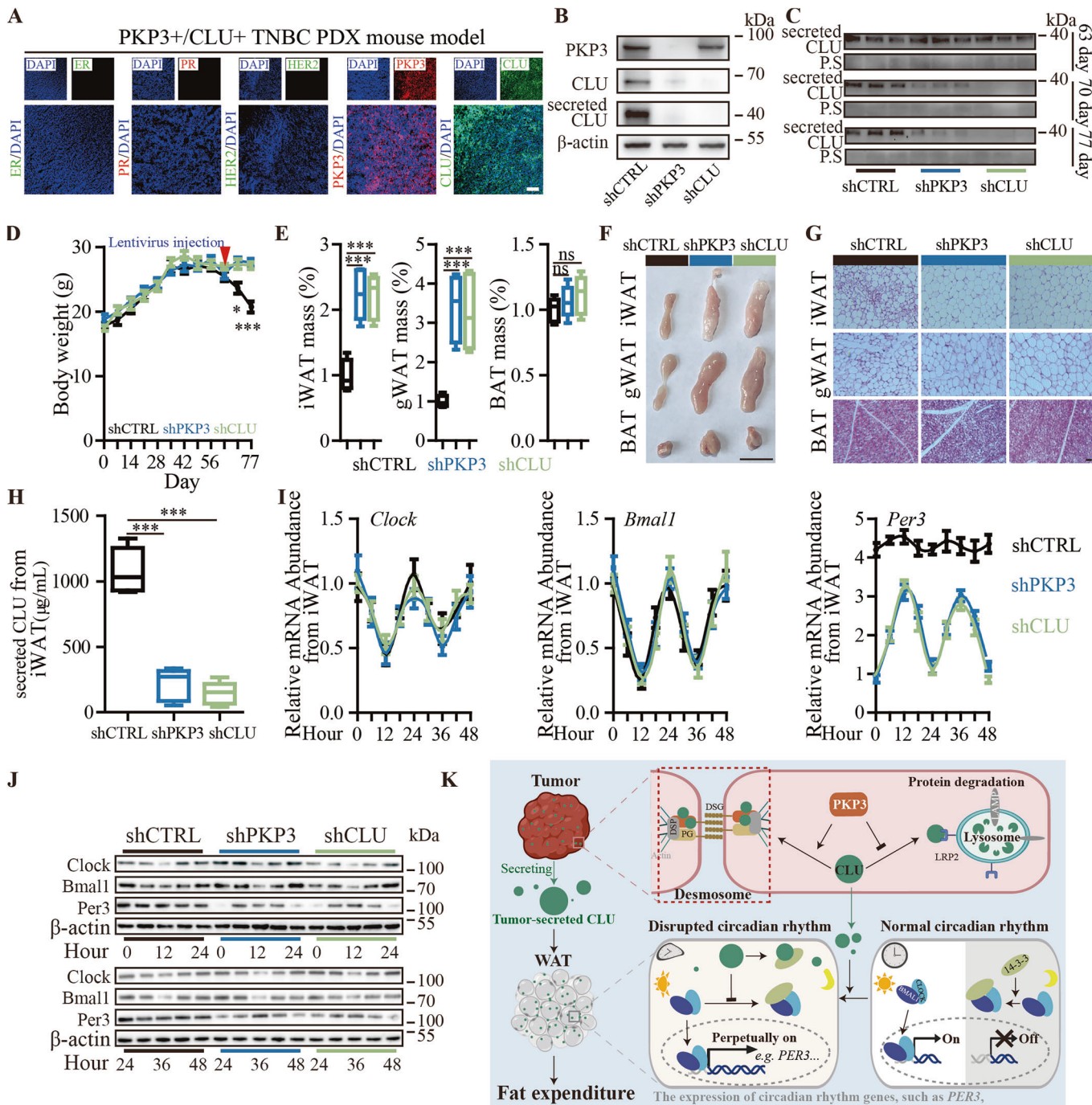

To investigate this point, we measured the expression of circadian rhythm genes, including *CLOCK*, *BMAL1* and *PER3*, in white fat derived from TNBC patients with cachexia by RT-qPCR. The results showed that *PER3*, but not *CLOCK* and *BMAL1*, was significantly overexpressed in the WAT of TNBC patients with cachexia compared to healthy women (Fig. 5I–K). In the WAT of TNBC patients with cachexia, patients with CLU + TNBC tumors had significantly higher level of *PER3* expression than patients with CLU- TNBC tumors (Fig. 5L). Furthermore, WAT's *PER3* mRNA levels were significantly positively correlated with the serum secreted CLU in TNBC patients with cachexia (Fig. 5M). These

results led us to claim that the disruption of the circadian rhythm in WAT by the tumor PKP3-CLU axis, which produces secreted CLU, is an important mechanism underlying the development of cachexia fat mass loss in advanced TNBC patients.

## Targeting the tumor PKP3-CLU axis effectively inhibits cachectic fat mass loss in vivo

There are no effective therapies against cancer-associated cachectic fat mass loss. To further prove that targeting the tumor PKP3-CLU axis is a potential strategy for the treatment of cachectic fat mass

◀ 

**Figure 6. Targeting the PKP3-CLU axis for cachectic fat mass loss in preclinical PDX TNBC mice.**

(A) Representative immunofluorescence images of immunohistochemistry assays showing the expression of ER, PR, HER2, PKP3 and CLU proteins in tumors collected from preclinical cachexia TNBC PDX mice before treatment, Scale bar: 100 μM. (B–J) Cachexia TNBC PDX mice with shPKP3 or shCLU treatment compared to the control: (B) Representative Western blot images showing the PKP3, CLU and secreted CLU levels in tumors; (C) Representative Western blot images showing tumor-secreted CLU levels in sera; (D) Body weight (shCTRL vs shPKP3 & shCLU: 70-day, $P < 0.0123$; 77-day, $P < 0.0003$; $n = 5$ per group); (E) Change of iWAT, gWAT, BAT weight (iWAT: shCTRL vs shPKP3, $P = 0.0003$; shCTRL vs shCLU, $P = 0.0002$; gWAT: shCTRL vs shPKP3, $P = 0.0003$; shCTRL vs shCLU, $P = 0.0008$; BAT: shCTRL vs shPKP3, $P = 0.3740$; shCTRL vs shCLU, $P = 0.1762$; $n = 5$ per group), in a box-plot, the center line represents the median of the data, while the lower and upper limits of the box correspond to the first quartile and third quartile, respectively; (F) Representative iWAT, gWAT, BAT photos, Scale bar: 1 cm; (G) Representative H&E staining images of iWAT, gWAT, BAT, Scale bar: 50 μM; (H) Tumor-secreted CLU levels in iWAT by ELISA assay (shCTRL vs shPKP3, $P < 0.0001$; shCTRL vs shCLU, $P < 0.0001$; $n = 5$ per group), in a box-plot, the center line represents the median of the data, while the lower and upper limits of the box correspond to the first quartile and third quartile, respectively; (I, J) CLOCK, BMAL1, PER3 expression in a 48-hour cycle by (I) RT-qPCR ($n = 4$ per group) and (J) Western blot assay. (K) A mechanistic model. The data are presented as mean ± SEM ($n = $ [X] biologically independent samples). *$p < 0.05$, **$p < 0.01$, ***$p < 0.001$, ns for not significant. Statistical comparisons between groups were performed using unpaired two-tailed Student's t tests. Source data are available online for this figure.

loss in advanced TNBC patients, we generated PDX mouse models by using PKP3 + /CLU+ tumors derived from TNBC patients with cachexia (Fig. 6A). When the PDX mice showed signs of developing cancer cachexia by the criteria of significant body weight loss >5%, the mice were treated with CTRL, anti-PKP3, or anti-CLU shRNA lentivirus by intratumoral injection. Both anti-PKP3 and anti-CLU treatment effectively downregulated CLU in tumors and secreted CLU in sera compared to control (Fig. 6B,C). When we monitored the progression of cachexia development by measuring the weight loss after treatment, we found that both anti-PKP3 and anti-CLU significantly alleviated the weight loss compared to control CTRL (Fig. 6D). When we examined the fat mass in our PDX TNBC cachexia mouse models, we found that both anti-PKP3 and anti-CLU significantly increased the size and weight of WAT compared to the control (Fig. 6E,F). Both anti-PKP3 and anti-CLU have larger white adipocytes full of stored fat compared to the control (Figs. 6G and EV5J). When we examined the level of tumor-secreted CLU in iWAT, the results showed that both anti-PKP3 and anti-CLU significantly decreased the level of secreted CLU in iWAT compared to the control (Fig. 6H). We measured the rhythmic mRNA and protein expression of circadian rhythm genes, including *Clock*, *Bmal1*, and *Per3*, in mouse iWAT at different time points in a 48-hour cycle by RT-qPCR and Western blot. RT-qPCR and Western blot results consistently showed that *Clock* and *Bmal1* exhibited normal rhythmic expression in iWAT of mice bearing TNBC tumors with and without anti-PKP3 and anti-CLU treatment (Fig. 6I,J). In comparison, anti-PKP3 and anti-CLU treatment, but not control CTRL treatment, restored the rhythmic expression of *Per3* in mouse iWAT (Fig. 6I,J). Taken together, these results establish an important role for the TNBC tumor PKP3-CLU axis and its produced tumor-secreted CLU in disrupting circadian rhythm and inducing cachectic fat mass loss, highlighting them as potential diagnostic markers and therapeutic targets for TNBC-associated cachectic fat mass loss (Fig. 6K).

## Discussion

Cachexia is a progressive metabolic disorder in patients with malignant or non-malignant diseases, including infectious diseases, such as AIDS and tuberculosis, and chronic diseases, such as congestive heart failure, chronic obstructive pulmonary disease, and multiple sclerosis, in addition to various cancers (Tisdale, 2009). In cancer patients, cachexia is characterized by systemic inflammation, body weight loss, adipose tissue atrophy, and skeletal

muscle wasting, with WAT atrophy being one of the hallmarks of cachexia (Fearon et al, 2012b; Schmidt et al, 2018). However, how tumors induce cachectic fat mass loss is not fully understood, and there are currently no effective therapies for cachectic fat mass loss. Therefore, there is an urgent need to further investigate the mechanism underlying cachectic fat mass loss and to identify novel therapeutic targets for the prevention and treatment of cachectic fat mass loss.

Here, using TNBC as a model, we demonstrated that the tumor-secreted CLU produced by PKP3-stabilized CLU in the tumor plays a key role in inducing cancer cachexia by entering the WAT to disrupt the circadian rhythm and cause cachectic fat mass loss in WAT. We identified and confirmed significantly higher levels of secreted CLU in sera of advanced TNBC patients with cachexia by MS and Elisa assy. In cachexia TNBC mouse models, tumor-secreted CLU entered into the circulation and then entered the WAT. In the WAT, tumor-secreted CLU competitively binds with 14-3-3 to inhibit its interaction with CLOCK/BMAL1, then pathologically promotes CLOCK/BMAL1 to continuously activate circadian rhythm-related genes, such as *Per3*, leading to misaligned circadian rhythm in WAT for white fat mass loss. In support, in a healthy non-cancerous mouse model without cancer, intravenous injection of purified secreted CLU disturbed circadian rhythm *Per3* expression in WAT and induced fat atrophy compared to control, where additional knockdown of *Per3* in the WAT of mice with secreted CLU treatment ameliorated the severity of fat mass loss in the WAT. CLU is a stress-activated chaperone, and its mature form can be secreted (Sanchez-Martin and Komatsu, 2020). We demonstrated that PKP3-stabilized CLU in the tumor desmosomes underlie the production of tumor-secreted CLU, where in tumor cells PKP3 stabilizes CLU by competitively binding with its lysosomal receptor LRP2 to increase and decrease CLU distribution in PKP3-coordinated desmosomal plaques and in lysosomes for protein degradation. Importantly, PKP3/CLU double-positive staining, increased secreted CLU and disturbed *PER3* expression ere observed in tumors, sera, and WAT samples of advanced TNBC patients with cachexia, respectively. And targeting the PKP3-CLU axis by PKP3 or CLU suppression effectively ameliorates cachexia in preclinical cachexia TNBC PDX mice. This study reveals PKP3-stabilized CLU in the tumor desmosomes and its release of tumor-secreted CLU into the circulation to disrupt circadian rhythm in WAT as the phenotypically related important mechanisms underlying the tumor-WAT communication for the induction of cachectic fat mass loss and cachexia in TNBC patients and highlights the PKP3-CLU axis in tumors, tumor-secreted CLU in

sera, and dysregulated circadian rhythm genes in WATs as promising biomarkers and therapeutic targets for cancer cachexia.

The circadian system is responsible for the internal functioning of the organism and regulates it according to environmental cues (Ruan et al, 2021). Circadian rhythm disorders have been associated with a wide range of diseases, from metabolic disorders to autoimmune diseases, as well as altered quality of life (Kulkarni et al, 2024). Dysregulated circadian rhythms have been associated with abnormalities in adipogenesis and lipid synthesis in non-cancerous settings. CLOCK inhibits adipogenesis and prevents fat accumulation (Sato and Sassone-Corsi, 2022). For example, ClockΔ19/Δ19 mice show weight gain associated with visceral fat accumulation. Clock suppression leads to the advancement of adipogenic differentiation and lipid synthesis at an early stage of differentiation in preadipocytes (Bass, 2024). In addition, Bmal1 promotes adipogenic differentiation and lipogenesis. Inhibition of Bmal1 expression in preadipocytes prevented adipogenesis and reduced the expression of adipogenic genes, such as PPARγ, adipocyte protein 2 (aP2), and C/EBP (Shimba et al, 2005). Meanwhile, overexpression of Bmal1 increased the mRNA levels of adipogenic enzymes and promoted lipogenesis (Zhang et al, 2014a). Mice with REV-ERBα detection, another clock gene, show disturbances in lipid metabolism leading to increased hepatic triglycerides and free fatty acids. Mice with Cry1 deletion gain less weight and accumulate less fat on a high-fat diet (Ma et al, 2021). The Per3 gene was found to exhibit strong circadian oscillations in preadipocytes, and deletion of the Per3 gene significantly increased adipogenesis in vivo (Aggarwal et al, 2017). However, whether and how circadian rhythm contributes to cancer cachexia remains unknown. Our results show that tumor-secreted CLU enters the WAT to induce cachexia by competitively binding with 14-3-3 to inhibit its interaction with circadian CLOCK/BMAL1 for pathological promoting CLOCK/BMAL1 in continuously activating circadian rhythm-related genes, including *PER3*. Thus, this study reveals a novel mechanistic link between circadian rhythm and cancer cachexia by demonstrating the important role of tumor-secreted CLU in disrupting circadian rhythm and promoting cachectic fat mass loss in WAT.

In cancer-related cachexia, tumor-secreted humoral cytokines play a crucial role in inter-organ/tissue communication (Yeom et al, 2021). Systemic coordination and integration between organs and tissues through inter-organ communication is important under normal conditions as well as in pathological conditions such as cancer and metabolic syndrome (Gonzalez et al, 2018). The wasting phenotype occurs mainly in WAT and skeletal muscle, and previous studies suggest that tumors induce inflammatory responses and metabolic dysfunction in fat and muscle by activating or secreting tumor-derived pro-inflammatory cytokines, such as tumor necrosis factor-α (TNF-α), interleukin-1α, interleukin-1β, IL-6, and transforming growth factor-β (Breen et al, 2020; Fearon et al, 2013; Hsu et al, 2017; Kir et al, 2016; Kir et al, 2014; Olson et al, 2021; Petruzzelli and Wagner, 2016; Tsai et al, 2018; Wang et al, 2018). Previous studies have shown various important roles of CLU in the regulation of tumor growth and cancer progression (Liu et al, 2021; Mamun et al, 2024). Some studies suggest that CLU may be involved in the regulation of inflammation. For example, CLU acts as a chemoattractant to promote directional migration of macrophages, a process that stimulates the expression and secretion of TNF-α and various chemotactic cytokines (Kang et al, 2014; Mamun et al, 2024). Previous studies have reported that CLU

participates in feeding behavior or diet-induced obesity, which contextualize its role in adipose tissue (Gil et al, 2013; Park et al, 2020). In addition, a daily rhythm controls the synthesis and release of cytokines, chemokines and cytolytic factors (Labrecque and Cermakian, 2015). Therefore, it would be interesting to investigate whether and how inflammatory cytokines are involved in the tumor (secreted) CLU-regulated fat mass loss and cancer cachexia. Besides TNBC, previous studies have shown that CLU are elevated in many other cancers, including prostate, lung, liver, colon, bladder, and pancreatic cancer, and high levels of serum secreted CLU are associated with poor prognosis in prostate, colon, esophageal, and invasive breast cancer (Zhang et al, 2023). Further work is needed to investigate whether and how tumor-secreted CLU regulates cachectic fat mass loss in WAT in other cancers beyond TNBC and the efficacy of CLU targeting therapy combined with other therapies, including chimeric antigen receptor T-cell immunotherapy (Liu et al, 2024a; Liu et al, 2024b; Liu et al, 2025). Nevertheless, these data suggest that cachectic fat mass loss induced by tumor-secreted CLU may be a general mechanism in wasting cancer patients with elevated tumor CLU.

In conclusion, our data demonstrate that the stabilization of CLU by PKP3 in tumors is a key axis underlying fat mass loss in cancer cachexia through secreted CLU into WAT to disrupt its circadian rhythm. Specific biomarkers for the diagnosis or therapeutic targets for the treatment of cancer cachexia remain unavailable (Yeom and Yu, 2022). Importantly, our data showed that the PKP3-CLU axis, serum CLU and key clock genes are clinically relevant for fat mass loss and cachexia in advanced TNBC patients, and targeting the PKP3-CLU axis effectively restores the circadian rhythm in WAT to ameliorate cachectic fat mass loss and weight loss in the preclinical TNBC PDX cachexia mouse model. Thus, this study establishes important roles for tumor (secreted) CLU and its upstream/downstream in the induction of cachectic fat mass loss and cancer cachexia and highlights them as promising targets for the future development of the diagnostic and therapeutic strategies for cancer cachexia.

# Methods

**Reagents and tools table**

| Reagent/Resource | Reference or Source | Identifier or Catalog Number |
|---|---|---|
| **Experimental models** | | |
| Balb/c-nude | Aniphe BioLab | NM10004 |
| 293T/17 | ATCC | ATCC® CRL-11268™ |
| 293T | ATCC | CRL-3216 |
| MDA-MB-468 | ATCC | ATCC® HTB-132™ |
| MDA-MB-231 | ATCC | ATCC® HTB-26™ |
| DH5α | TakaRa | 9057 |
| **Recombinant DNA** | | |
| pCMV3-Clusterin-HA | SinoBiological | HG11297-CY |
| pCMV3-PKP3 | SinoBiological | HG19874-UT |

| Reagent/Resource | Reference or Source | Identifier or Catalog Number |
|---|---|---|
| **Antibodies** | | |
| Anti-CLU | Cell Signaling Technology | 42143S |
| Anti-PKP3 | Invitrogen | 35-7600 |
| Anti-CLOCK | ABclonal Technology | A5633 |
| Anti-BMAL1 | ABclonal Technology | A4714 |
| Anti-PER3 | ABclonal Technology | A2219 |
| Anti-14-3-3 | ABclonal Technology | A23829 |
| Anti-DSG1 | ABclonal Technology | A9812 |
| Anti-LAMP1 | Cell Signaling Technology | 15665 |
| Anti-FLAG | Sigma | F7425 |
| Anti-HA | ABclonal Technology | AE036 |
| Anti-$\beta$-actin | Cell Signaling Technology | 3700 |
| Anti-ER | Cell Signaling Technology | 13258 |
| Anti-PR | Cell Signaling Technology | 3172 |
| Anti-HER2 | Cell Signaling Technology | 2165 |
| Rabbit IgG | Cell Signaling Technology | 2729 |
| Mouse IgG | Cell Signaling Technology | 5415 |
| Goat anti-Rabbit IgG (H + L) Cross-Adsorbed Secondary Antibody, Alexa Fluor™ 488 | Invitrogen | A-11008 |
| Goat anti-Mouse IgG (H + L) Cross-Adsorbed Secondary Antibody, Alexa Fluor™ 488 | Invitrogen | A-11001 |
| Goat anti-Rabbit IgG (H + L) Cross-Adsorbed Secondary Antibody, Alexa Fluor™ 555 | Invitrogen | A-21428 |
| Goat anti-Mouse IgG (H + L) Cross-Adsorbed Secondary Antibody, Alexa Fluor™ 555 | Invitrogen | A-21422 |
| **Oligonucleotides and other sequence-based reagents** | | |
| shCTRL | TAAGGCTATGAAGAGATAC | |
| shPKP3 | CATCTACGACAACGCTGACAA | |
| shCLU | GCTAAAGTCCTACCAGTGGAA | |
| **Chemicals, enzymes and other reagents** | | |
| Fetal Bovine Serum | WISENT | 086-150 |
| Penicillin-Streptomycin | WISENT | 450-201-EL |
| PBS | WISENT | 311-011-CL |
| DMEM | WISENT | 319-005-CL |
| TRYPSIN | WISENT | 325-043-CL |
| Cycloheximide | MedChemExpress | HY-12320 |
| MG-132 | MedChemExpress | HY-13259 |
| Oil Red O | Sigma-Aldrich | O0625 |
| Hematoxylin and Eosin Staining Kit | Solarbio | G1120-100 |

| Reagent/Resource | Reference or Source | Identifier or Catalog Number |
|---|---|---|
| RNA isolater Total RNA Extraction Reagent | Vazyme | R401-01 |
| SteadyPure RNA Extraction Kit | Accurate Biology | AG21024 |
| ChamQ SYBR qPCR Master Mix | Vazyme | Q341-02 |
| HiScript Q RT SuperMix for qPCR | Vazyme | R122-01 |
| One-Step PAGE Gel Fast Preparation Kit | Vazyme | E303-01 |
| BSA | Biosharp | BS114-250g |
| RIPA Lysis Buffer | Beyotime | P0013D |
| PMSF | Beyotime | ST505 |
| Tris | BioFroxx | 1115GR500 |
| TWEEN-20 | Sinopharm | 30189328 |
| Hydrochloric acid | Sinopharm | 10011018 |
| SDS | Sinopharm | 30166428 |
| NaCl | Sinopharm | 10019318 |
| KCl | Sinopharm | 10016318 |
| **Software** | | |
| Graphpad Prism | Graphpadsoftwave | Graphpad.com |
| ImageJ | National Institutes of Health | |
| **Other** | | |
| Cell culture incubator (Forma™ 3 Series Water-Jacketed $CO_2$ Incubator, 184 L) | Thermo Fisher Scientific, USA | |
| Biological safety cabinet (Class II, Type A2 Biological Safety Cabinet) | Thermo Fisher Scientific, USA | |
| StepOnePlus™ Real-Time PCR system | Bio-Rad Laboratories, Inc., USA | |
| Azure multi-functional molecular imaging system | Azure Biosystems, USA | |
| Laser scanning confocal microscope (Nikon AX) | Nikon Corporation, Japan | |

## Cell lines and culture condition

MDA-MB-468, MDA-MB-231, 293 T/17 and 293 T were purchased from the American Type Culture Collection. These cells were maintained in Dulbecco's modified essential medium supplemented with 10% heat-inactivated fetal bovine serum and 1% penicillin-streptomycin solution in a humidified incubator at 37 °C with 5% $CO_2$.

## Real-Time PCR (RT-qPCR)

In brief, RNeasy kit (OMEGA) was used to extract total RNA from the treated cells, PrimeScript RT reagent kit (TaKaRa) was used to synthesize cDNAs, then real-time PCR reaction was carried out

with the SYBR Premix Ex Taq (TaKaRa). Real-time PCR primer sequences were listed in Dataset EV5.

## Western blotting and Co-immunoprecipitation (Co-IP) assay

Western blotting was performed according to the standard method as aforementioned (Liu et al, 2021). Co-IP assay was performed using related antibody according to the standard method as aforementioned (Liu et al, 2021). In brief, Primary antibody-conjugated beads were incubated with cell lysates at 4 °C for 12 h. Immunoprecipitated proteins were eluted from beads. The immunoprecipitated proteins were analyzed by mass spectrum or Western blot. CLU protein was obtained from MedChemExpress. The catalog numbers of all our antibodies were as follows: CLU (Cell Signaling Technology, 42143S), PKP3 (Invitrogen, 35-7600), CLOCK (ABclonal Technology, A5633), BMAL1 (ABclonal Technology, A4714), PER3 (ABclonal Technology, A2219), 14-3-3 (ABclonal Technology, A23829), DSG1 (ABclonal Technology, A9812), LAMP1 (Cell Signaling Technology, 15665), FLAG (Sigma, F7425), HA (ABclonal Technology, AE036), β-actin (Cell Signaling Technology, 3700), Rabbit IgG (Cell Signaling Technology, 2729), and Mouse IgG (Cell Signaling Technology, 5415).

## Mass spectrum (MS) analysis

Briefly, the protein samples from serum or Co-IP assay were reduced and alkylated, and then enzymatically digested by trypsin at 37 °C for 20 h. The enzyme digestion product was desalted, lyophilized and re-dissolved in 0.1% FA solution. After the column was equilibrated with 95% of 0.1% formic acid in water, the sample was loaded onto the Trap column by an autosampler. Mass-to-charge ratios of peptides and fragments of peptides were collected according to the following method: 20 fragment profiles were collected after each full scan. The raw file of the mass spectrometry test was analyzed by Proteome Discoverer 1.4 software in the corresponding databases, and the identified proteins were finally obtained.

## Transcriptome sequencing

Briefly, mRNA library constructing and sequencing was performed by Azenta Life Science. Differential expression analysis was performed by using DESeq2 Bioconductor package. Padj of genes were setted <=0.05 to detect differential expressed ones. GOSeq (v1.34.1) was used identifying Gene Ontology (GO) terms that annotate a list of enriched genes with a significant padj less or equal than 0.05. And topGO was used to plot DAG.

## Protein stability analysis

Briefly, to carry out cycloheximide (CHX) chase assay, CHX (50 μg/mL) was used to treat cells as indicated time. To inhibit UPS or ALP, UPS inhibitor MG132 (10 μm) or ALP inhibitor CQ (10 μm) was used to treat cells for 24 h. Then, the lysates of the treated cells were analyzed by Western blotting.

## Immunofluorescence assay

In brief, cells were growing on glass slides with various density in 24-well plates for 24 h. The cells were fixed with paraformaldehyde

(4%, 1 h) and permeabilized/blocked with PBS (1 h) which contained Triton X-100 (0.1%) and BSA (0.1%). Anti-PKP3 (Invitrogen, 35-7600), anti-DSG1 (ABclonal Technology, A9812), anti-CLU (Cell Signaling Technology, 42143S) and anti-LAMP1 (Cell Signaling Technology, 15665) and the fluorescent secondary antibodies were used to detect protein in cells as aforementioned.

## Lentiviral shRNA knockdown and gene overexpression

Lentiviral shRNA knockdown was used to generate stable cells with PKP3 and CLU knocked down. Briefly, to obtain PKP3, CLU stable suppression cell lines, related lentiviral shRNAs were cloned into pLKO.1 lentiviral vector with EcoRI and AgeI restriction enzymes. The sequences were shown as follows: shPKP3: 5'-CATCTACGA-CAACGCTGACAA-3'; 5'-CCTGTGTGCATCTTTGAGGGT-3'; shCLU:5'-GCTAAAGTCCTACCAGTGGAA-3'; 5'- CAGGGAAG-TAAGTACGTCAAT-3'; shCTRL: 5'-TAAGGCTATGAAGAGA-TAC-3'. Lentiviral vector and packaging vector mix were transfected into 293 T/17. The produced viruses were transfected into MDA-MB-468 and MDA-MB-231. The cells were cultured with 2 μg/mL puromycin for at least 7 days. Lentiviral gene overexpression was used to generate stable cells with CLU overexpressed. Briefly, to overexpress CLU in PKP3 suppression cell line, the sequence of CLU was cloned into pLVX-Puro lentiviral vector. The cell construction process was mentioned above. The expression of genes in the stable cells was confirmed to be successfully knocked down by RT–qPCR and western blotting.

## Cell line-derived cachectic xenograft mouse model

Briefly, 3–4 weeks female nude mice were obtained from Nanjing University's model animal research center. constructed cells (5 × $10^6$) were injected into the mammary fat pads of each mouse to generate cell line-derived xenograft mouse model. Tumor size and body weight were monitored every day. Mice developed cachexia when body weight loss >5%. All the mice were housed with a standard chow diet on a 12 h light/12 h dark cycle. The blood was obtained from the tail vein of mice for further investigated. The WAT obtained from mice directly frozen for further studies. All mouse experiments were conducted in compliance with the guidelines approved by the Animal Ethics and Welfare Committee of Nanjing Normal University.

## Energy metabolism measurement

Briefly, tumor-bearing mice were individually housed in metabolic cages (PhenoMaster) of Nanjing Advanced Academy of Life and Health. $O_2$ consumption, $CO_2$ production and heat were monitored over a 48-h period. Data were collected and analyzed using MetaScreen-Data Collection Software.

## Clinical samples collection

Tumor, blood samples and WAT were collected from 20 female TNBC patients with cachexia with body mass index (<18.5) and 21 female TNBC patients without cachexia with body mass index (18.5-24). Healthy blood samples were obtained from 22 healthy

female persons with body mass index (18.5–24) and from 21 healthy female person with body mass index (<18.5). The clinical sample collection and experiment were carried out in accordance with guidelines and protocols approved by the Ethics and Scientific Committees of First Affiliated Hospital of Soochow University or Affiliated Cancer Hospital of Nanjing Medical University. All the human participants signed a letter of authorization. All samples, images and clinical data were reviewed and interpreted by experienced pathologists.

## Immunohistochemistry and hematoxylin-eosin staining assay

Immunohistochemistry assay and hematoxylin-eosin staining assay was performed as forementioned (Liu et al, 2022). In brief, according to the intensity of ER (Cell Signaling Technology, 13258), PR (Cell Signaling Technology, 3172), HER2 (Cell Signaling Technology, 2165), PKP3 (Invitrogen, 35-7600), CLU (Cell Signaling Technology, 42143S), brown, buffy, light yellow, and no pigmentation are scored 3, 2, 1, and 0, respectively. The percentage of pigmented cells as follows: >75%: 4, 51–75%: 3, 26–50%: 2, <5%: 0. The final score was calculated by multiplying above two scores: 9–12 scores ($+ + +$), 5–8 scores ($+ +$), 3–4 scores ($+$), 0–2 scores ($-$). Hematoxylin-eosin staining was used to stain nucleus and cytoplasm, respectively.

## Enzyme-linked immunosorbent assay (ELISA)

Briefly, enzyme-linked immunosorbent assay was performed using commercial ELISA kit (Abcam) according to standard method. The fat was weighted and lysis to detect CLU expression using related ELISA kit (ab174447). Blood samples were centrifuged; the supernatant was used for testing using related ELISA kit.

## Patient-derived cachectic xenograft (PDX) mouse model

Breast tumors were obtained from First Affiliated Hospital of Soochow University. Breast tumors were grafted into the mammary fat pads of each 3–4 weeks female NSG mice. Tumor size and body weight were monitored every day. Mice developed cachexia when body weight loss >5%. All the mice were housed with a standard chow diet on a 12 h light/12 h dark cycle. The concentrated lentivirus was directly injected into the tumor. The blood was obtained from the tail vein of mice for further investigated. The WAT obtained from mice directly frozen for further studies. Immunofluorescence assay was performed to detect ER, PR, HER2, PKP3, CLU expression in the PDX-tumor as abovementioned. The clinical sample collection and experiment were carried out in accordance with guidelines and protocols approved by the Ethics and Scientific Committees of First Affiliated Hospital of Soochow University, and all mouse experiments were conducted in compliance with the guidelines approved by the Animal Ethics and Welfare Committee of Nanjing Normal University.

## Statistical analysis

All data were analyzed via GraphPad Prism software and expressed as mean ± standard error. Statistical quantitative analysis was performed by t-test.

## Data availability

All the data was presented in this article and Supplementary figures and Datafiles. RNA-seq data have been deposited into the short reads archive database of National Center for Biotechnology Information (https://dataview.ncbi.nlm.nih.gov) with the identifier (BioProject: PRJNA1287256).

The source data of this paper are collected in the following database record: biostudies:S-SCDT-10_1038-S44318-025-00661-4.

## Peer review information

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

## Acknowledgements

This study was funded by the National Natural Science Foundation of China (Grant No.: 82103099 & 82372991) and Jiangsu Provincial Medical Innovation Center (Grant No. CXZX202224).

## Author contributions

**Yan Liu**: Conceptualization; Data curation; Formal analysis; Funding acquisition; Investigation; Methodology; Writing—original draft; Writing—review and editing. **Yehui Zhou**: Data curation; Formal analysis; Investigation; Methodology. **Mengmeng Zhang**: Investigation. **Jin Zhang**: Investigation. **Jiahui Chen**: Investigation. **Long Chen**: Investigation. **Jia Tian**: Investigation. **Xiang Lv**: Data curation; Investigation; Methodology. **Xinxing Ma**: Investigation. **Jing Xu**: Investigation. **Jingwei Shi**: Methodology. **Liming Chen**: Conceptualization; Supervision; Funding acquisition; Writing—review and editing.

Source data underlying figure panels in this paper may have individual authorship assigned. Where available, figure panel/source data authorship is listed in the following database record: biostudies:S-SCDT-10_1038-S44318-025-00661-4.

## Disclosure and competing interests statement

The authors declare no competing interests.

# Expanded View Figures

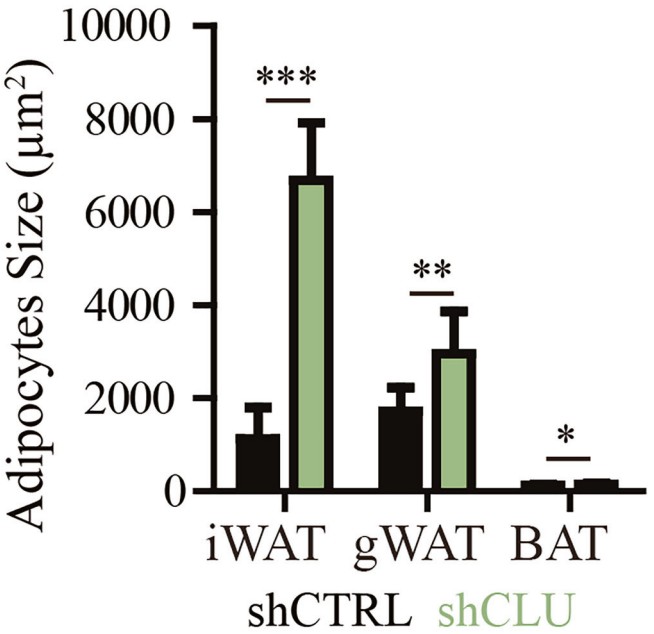

**Figure EV1.**   **Related to** Fig. 1. Statistical analysis of H&E staining of iWAT, gWAT, BAT in Fig. 1G (shCTRL vs shCLU: iWAT, $P < 0.0001$; gWAT, $P = 0.0014$; BAT, $P = 0.0229$; $n = 10$ per group). The data are presented as mean ± SEM. *$p < 0.05$, **$p < 0.01$, ***$p < 0.001$. Statistical comparisons between groups were performed using unpaired two-tailed Student's t tests. Source data are available online for this figure.

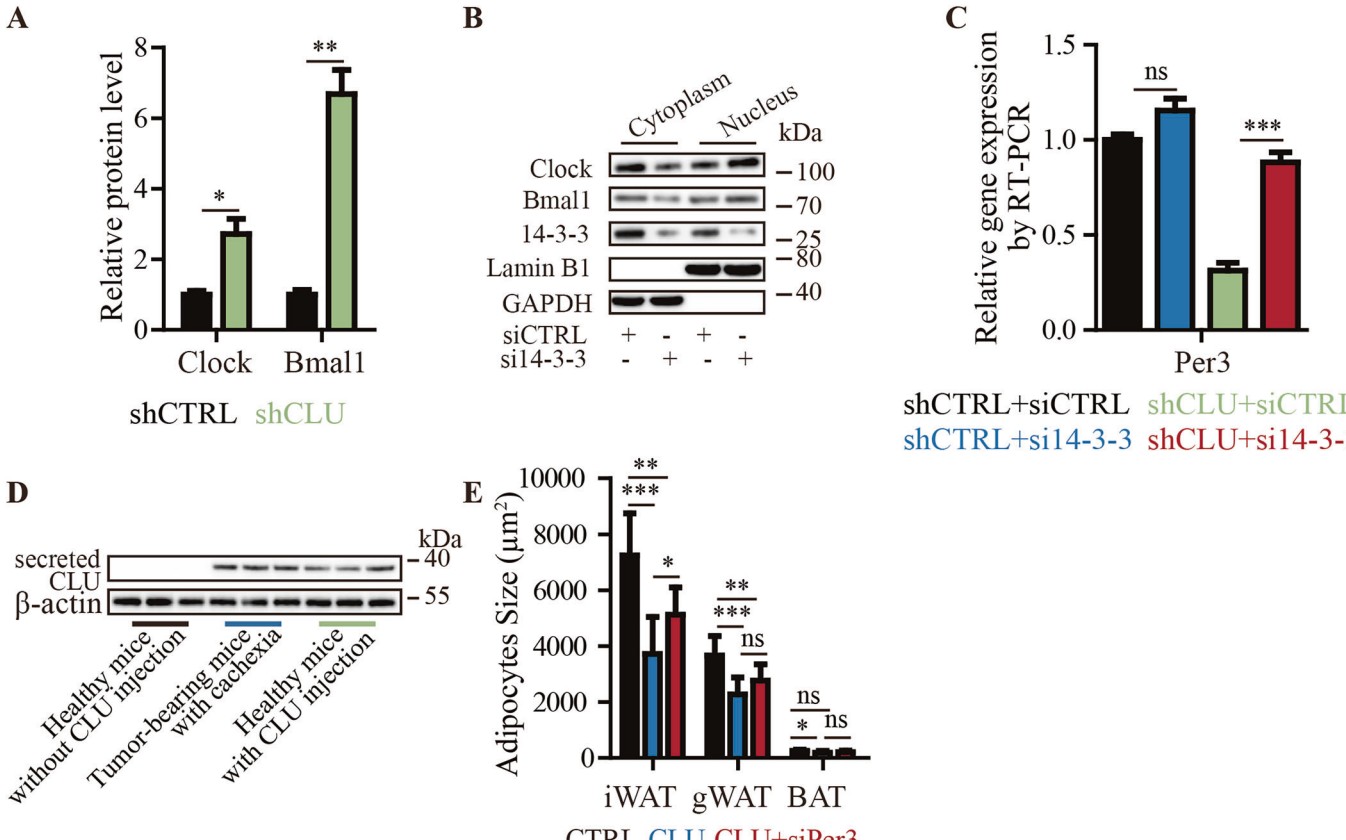

**Figure EV2.** **Related to** Fig. 2. **(A)** Quantification of western blot by Co-IP assays in Fig. 2F (shCTRL vs shCLU: Clock, $P = 0.0181$; Bmal1, $P = 0.0012$; $n = 3$ per group). **(B)** Representative Western blot images of a nucleocytoplasmic separation assay showing the levels of cytoplasmic and nuclear Clock, Bmal1, and 14-3-3 in iWAT cells with and without 14-3-3 knockdown in mice bearing CLU knockdown tumors. **(C)** Per3 expression in iWAT cells with and without 14-3-3 knockdown by RT-qPCR. The shCLU and shCTRL stand mice bearing tumor with and without CLU knockdown, respectively (shCTRL+siCTRL vs shCTRL+si14-3-3: *Per3*, $P = 0.0691$; shCLU+siCTRL vs shCLU +si14-3-3: *Per3*, $P = 0.0001$; $n = 4$ per group). **(D)** circulating CLU levels in iWAT of mice by western blot. **(E)** Statistical analysis of H&E staining of iWAT, gWAT, BAT in Fig. 2L (iWAT: CTRL vs CLU, $P < 0.0001$; CTRL vs CLU+siPer3, $P = 0.0016$; CLU vs CLU+siPer3, $P = 0.0151$; gWAT: CTRL vs CLU, $P = 0.0001$; CTRL vs CLU+siPer3, $P = 0.0057$; CLU vs CLU+siPer3, $P = 0.0819$; BAT: CTRL vs CLU, $P = 0.0137$; CTRL vs CLU+siPer3, $P = 0.0511$; CLU vs CLU+siPer3, $P = 0.5122$; $n = 10$ per group). The data are presented as mean ± SEM. *$p < 0.05$, **$p < 0.01$, ***$p < 0.001$, ns for not significant. Statistical comparisons between groups were performed using unpaired two-tailed Student's t tests. Source data are available online for this figure.

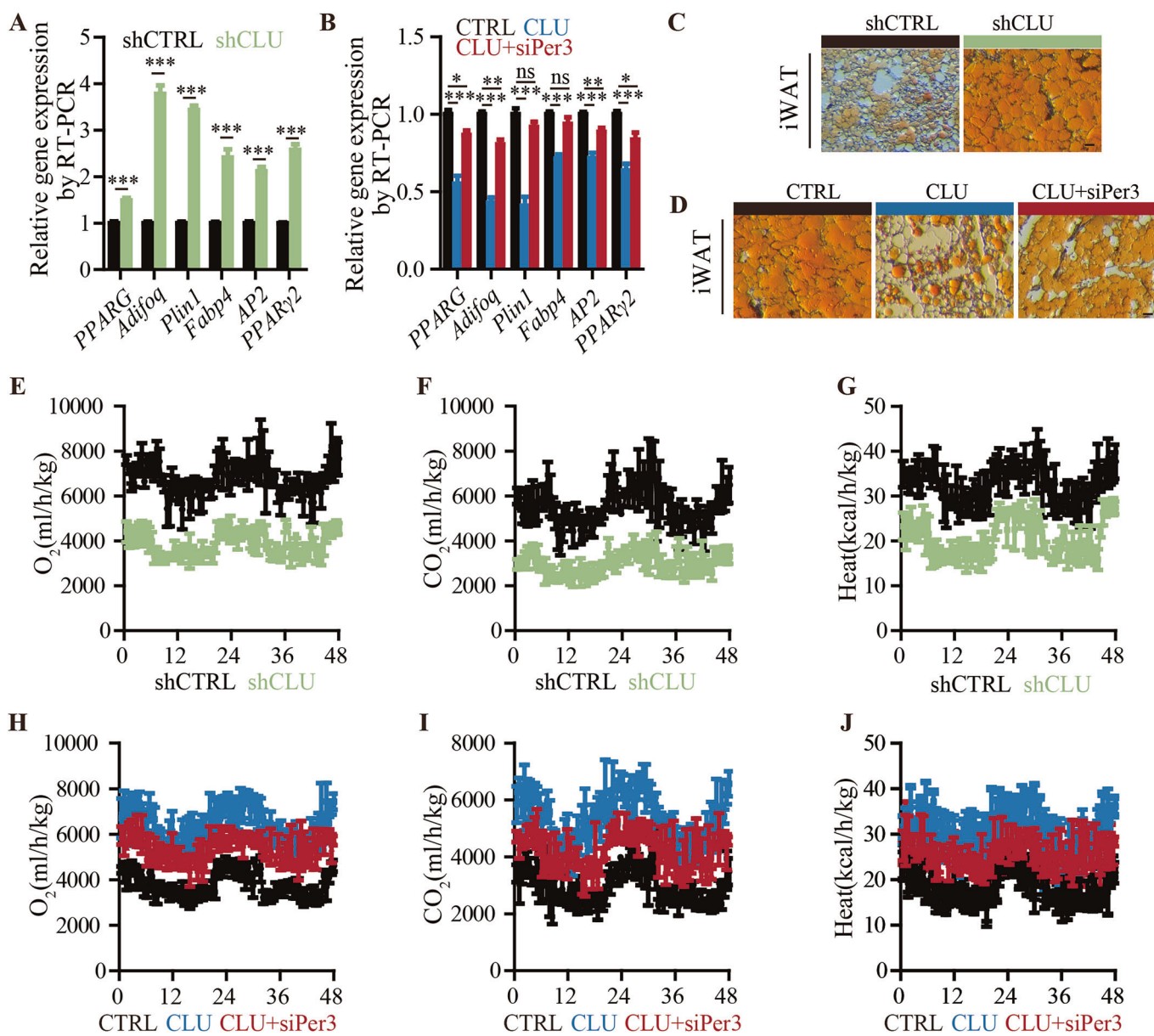

**Figure EV3. The CLU-Per3 axis contribute to adipogenesis and energy expenditure.**

(A) Adipogenesis-related gene expression by RT-qPCR in iWAT of MDA-MB-468 tumor-bearing mice with indicated treatment (shCTRL vs shCLU: *PPARG*, $P = 0.0003$; *Adifoq*, $P < 0.0001$; *Plin1*, $P < 0.0001$; *Fabp4*, $P = 0.0002$; *AP2*, $P < 0.0001$; *PPARγ2*, $P < 0.0001$; $n = 4$ per group). (B) Adipogenesis-related gene expression by RT-qPCR in iWAT of healthy nude mice with indicated treatment (CTRL vs CLU: *PPARG*, $P = 0.0002$; *Adifoq*, $P < 0.0001$; *Plin1*, $P = 0.0002$; *Fabp4*, $P = 0.0001$; *AP2*, $P = 0.0002$; *PPARγ2*, $P = 0.0003$; CTRL vs CLU+siPer3: *PPARG*, $P = 0.0142$; *Adifoq*, $P = 0.0018$; *Plin1*, $P = 0.1590$; *Fabp4*, $P = 0.2300$; *AP2*, $P = 0.0088$; *PPARγ2*, $P = 0.0163$; $n = 10$ per group). (C, D) Adipogenesis by using Oil Red O staining in iWAT of mice. (E, H) Oxygen consumption, (F, I) carbon dioxide generation and (G, J) heat production by metabolic cages in mice ($n = 3$ per group). The data are presented as mean ± SEM. *$p < 0.05$, **$p < 0.01$, ***$p < 0.001$, ns for not significant. Statistical comparisons between groups were performed using unpaired two-tailed Student's t tests. Source data are available online for this figure.

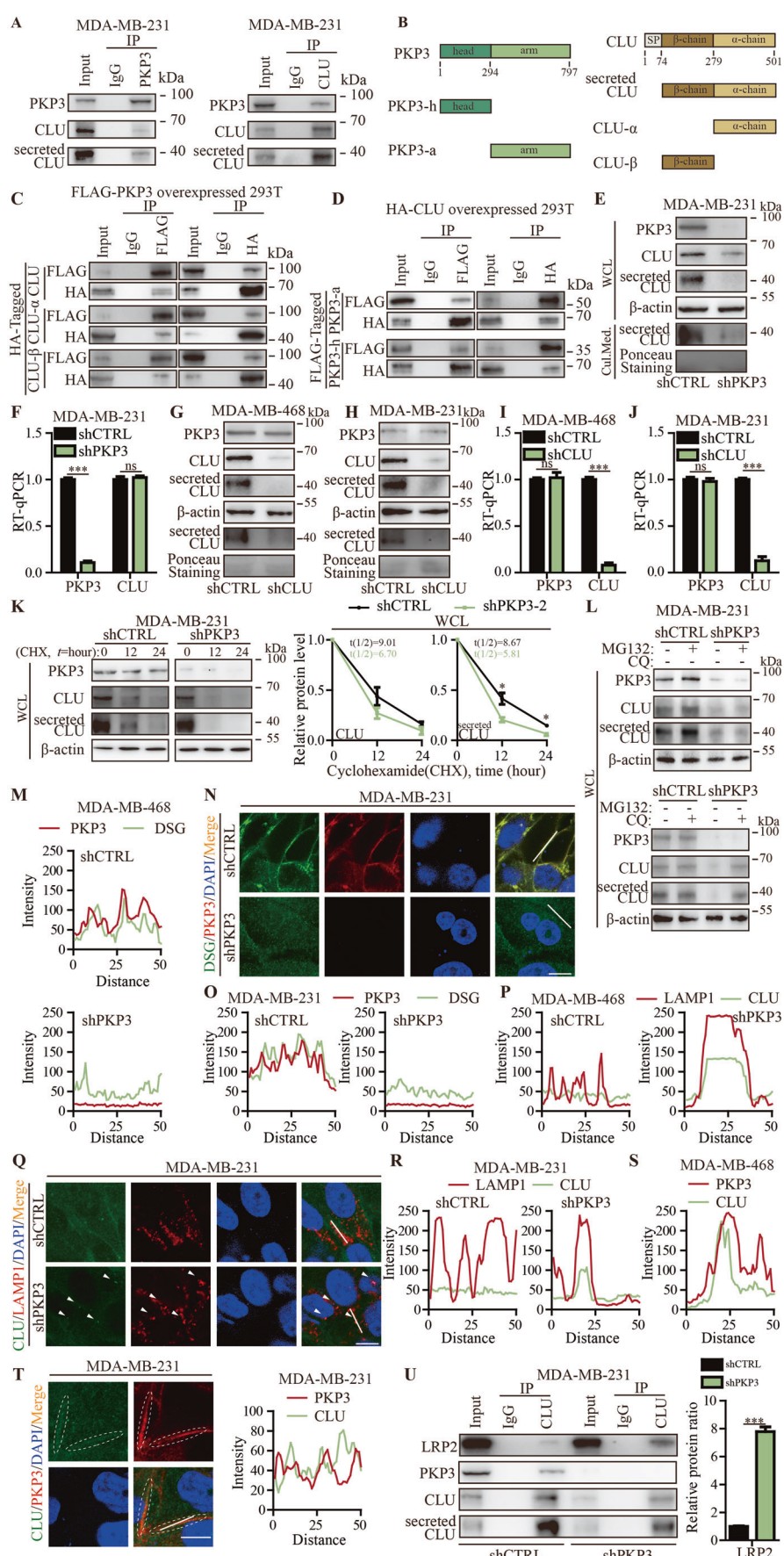

◀ **Figure EV4. CLU stabilization by PKP3 in TNBC cancer cells.**

(A) Representative Western blot images showing PKP3, CLU and secreted CLU levels in Co-IP assays using TNBC MDA-MB-231 cells. (B) Schematic representation of full-length PKP3, CLU, secreted CLU and their domain truncates. (C) Representative Western blot images of Co-IP assays using 293 T cells with ectopic overexpression of FLAG-PKP3 and HA-CLU or its domain truncates (HA-CLU-α, HA-CLU-β). (D) Representative Western blot images of Co-IP assays using 293 T cells with ectopic overexpression of HA-CLU and PKP3 truncates (FLAG-PKP3-a, FLAG-PKP3-h). (E) Representative Western blot images showing PKP3, CLU and secreted CLU protein levels in MDA-MB-231 cells and secreted CLU protein levels in cell culture medium (Cul. Med.) upon PKP3 knockdown compared to controls. (F) PKP3 and CLU mRNA levels in MDA-MB-231 cells by RT-qPCR. (G–J) PKP3 and CLU mRNA and PKP3, CLU and secreted CLU protein levels in TNBC MDA-MB-468 and MDA-MB-231 cancer cells in cell culture medium (Cul. Med.) upon CLU knockdown compared to controls. (G, H) Representative Western blot images. (I, J) RT-qPCR ($n = 4$ per group). (K) Representative Western blot images and their quantification of PKP3, CLU and secreted CLU protein levels in MDA-MB-231 cells upon PKP3 knockdown compared to control after CHX treatment (shCTRL vs shPKP3-2: secreted CLU, 12-hour, $P = 0.0261$; 24-hour, $P = 0.0137$). (L) Representative Western blot images showing PKP3, CLU and secreted CLU protein levels in cancer cells upon MG132 or CQ treatment in PKP3 knockdown MDA-MB-231 cells compared to control. (M) Colocalization analysis of Fig. 3F. (N) Representative immunofluorescence images and (O) colocalization analysis showing PKP3 and DSG localization in MDA-MB-231 cells after PKP3 knockdown compared to the control. Green for DSG, red for PKP3, blue for DAPI and orange for DSG-PKP3 colocalization. Scale bar: 10 μM. (P) Colocalization analysis of Fig. 3G. (Q) Representative immunofluorescence images and (R) colocalization analysis showing PKP3 and CLU localization in MDA-MB-231 cells. White dashed lines highlight the intercellular plaque regions, green for CLU, red for PKP3, blue for DAPI, orange for CLU-PKP3 colocalization. Scale bar: 10 μM. (S) Colocalization analysis of Fig. 3H. (T) Representative immunofluorescence images and colocalization analysis showing CLU and LAMP1 localization in MDA-MB-231 cells upon PKP3 knockdown compared to the control. Green for CLU, red for LAMP1, blue for DAPI and orange for CLU-LAMP1 colocalization. Scale bar: 10 μM. (U) Representative Western blot images and quantification of LRP2, PKP3, CLU and secreted CLU levels in Co-IP assays using MDA-MB-231 cells upon PKP3 knockdown compared to the control (shCTRL vs shPKP3: *LRP2*, $P < 0.0001$; $n = 3$ per group). The data are presented as mean ± SEM ($n = $ [X] biologically independent samples). *$p < 0.05$, ***$p < 0.001$, ns for not significant. Statistical comparisons between groups were performed using unpaired two-tailed Student's t tests. Source data are available online for this figure.

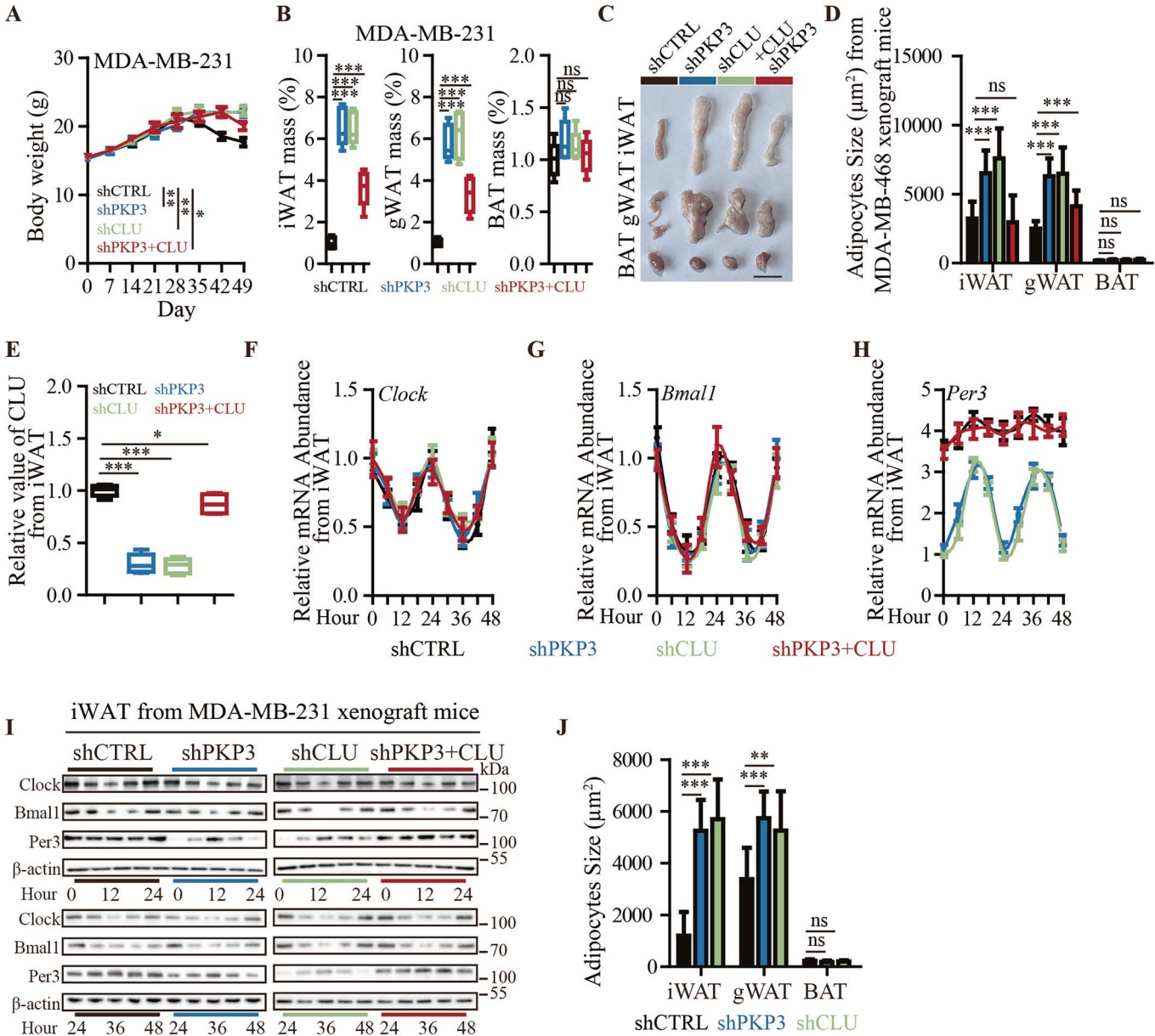

**Figure EV5. Disruption of WAT's circadian rhythm by tumor PKP3-CLU axis for cachectic fat mass loss.**

(A–I) MDA-MB-231 tumor-bearing mice with PKP3 knockdown, CLU knockdown, or PKP3 knockdown combined with ectopic overexpression of CLU ($n = 5$ per group): **(A)** Body weight (shCTRL vs shPKP3, $P = 0.0015$; shCTRL vs shCLU, $P = 0.0012$; shCTRL vs shPKP3+CLU, $P = 0.0260$; $n = 5$ per group); **(B)** Weight change of iWAT, gWAT, BAT (iWAT: shCTRL vs shPKP3, $P < 0.0001$; shCTRL vs shCLU, $P < 0.0001$; shCTRL vs shPKP3+CLU, $P = 0.0003$; gWAT: shCTRL vs shPKP3, $P < 0.0001$; shCTRL vs shCLU, $P < 0.0001$; shCTRL vs shPKP3+CLU, $P < 0.0001$; BAT: shCTRL vs shPKP3, $P = 0.1656$; shCTRL vs shCLU, $P = 0.1322$; shCTRL vs shPKP3+CLU, $P = 0.7618$; $n = 5$ per group), in a box-plot, the center line represents the median of the data, while the lower and upper limits of the box correspond to the first quartile and third quartile, respectively; **(C)** Representative iWAT, gWAT, BAT photos, Scale bar: 1 cm; **(D)** Statistical analysis of H&E staining of iWAT, gWAT, BAT in Fig. 4D (iWAT: shCTRL vs shPKP3, $P < 0.0001$; shCTRL vs shCLU, $P < 0.0001$; shCTRL vs shPKP3+CLU, $P = 0.7042$; gWAT: shCTRL vs shPKP3, $P < 0.0001$; shCTRL vs shCLU, $P < 0.0001$; shCTRL vs shPKP3+CLU, $P = 0.0009$; BAT: shCTRL vs shPKP3, $P = 0.2816$; shCTRL vs shCLU, $P = 0.1173$; shCTRL vs shPKP3+CLU, $P = 0.0980$; $n = 10$ per group). **(E)** Tumor-secreted CLU levels in iWAT by ELISA assay (shCTRL vs shPKP3, $P < 0.0001$; shCTRL vs shCLU, $P < 0.0001$; shCTRL vs shPKP3+CLU, $P = 0.0277$; $n = 5$ per group); **(F–I)** CLOCK, BMAL1, PER3 expression in WAT in a 48-hour cycle by (F–H) RT-qPCR ($n = 4$ per group) and (I) Western blot assay. **(J)** Statistical analysis of H&E staining of iWAT, gWAT, BAT in Fig. 6G (iWAT: shCTRL vs shPKP3, $P < 0.0001$; shCTRL vs shCLU, $P < 0.0001$; gWAT: shCTRL vs shPKP3, $P = 0.0002$; shCTRL vs shCLU, $P = 0.0068$; BAT: shCTRL vs shPKP3, $P = 0.0745$; shCTRL vs shCLU, $P = 0.1963$; $n = 3$ per group). The data are presented as mean ± SEM ($n = [X]$ biologically independent samples). *$p < 0.05$, **$p < 0.01$, ***$p < 0.001$, ns for not significant. Statistical comparisons between groups were performed using unpaired two-tailed Student's t tests. Source data are available online for this figure.

