## [Peer Review File · The EMBO Journal]

Tumor-secreted clusterin promotes cachectic fat wasting via disrupting circadian gene expression and adipogenesis

Yan Liu, Yehui Zhou, Jin Zhang, Mengmeng Zhang, Jiahui Chen, Long Chen, Jia Tian, Xiang Lv, Xinxing Ma, Jing Xu, Jingwei Shi, and Liming Chen

Corresponding authors: Yan Liu (llliuyan@sina.com) , Liming Chen (chenliming1981@u.nus.edu)

Review Timeline:

Submission Date:	20th Feb 25
Editorial Decision:	11th Apr 25
Revision Received:	10th Jul 25
Editorial Decision:	22nd Sep 25
Revision Received:	9th Oct 25
Accepted:	7th Nov 25

Editor: Daniel Klimmeck

Transaction Report:

Dear Dr Liu,

Thank you again for the submission of your manuscript (EMBOJ-2025-120556) to The EMBO Journal, as well as for your patience with our feedback at this time. As mentioned earlier, your study was assessed by three reviewers with expertise in tumor biology, circadian control and adipogenesis, whose comments are enclosed below.

As you will see from the experts' reports, the referees acknowledge the analysis and potential interest and value of your findings. However, they also express major concerns regarding completeness and in vivo relevance of the findings, which need to be addressed thoroughly to make them supportive of publication in the EMBO Journal. In more detail, reviewer #2 raises substantial issues related to proof of a distinctive cachexia- vs broader tumor-regulatory role of CLU (ref#2, standfirst, pt.4); s/he also points to insufficient insights into the adipocyte biology affected by CLU (ref#2, pts. 2,5). Referee #3 agrees in that the mechanistic details provided need to be strengthened (ref#3, standfirst). Further, the reviewers raise a number of issues related to the human control data sets, presentation of the findings, improved methods annotation required, statistics applied and overall discussion of related literature, that would need to be conclusively addressed to achieve the level of robustness and clarity needed for The EMBO Journal.

Given the overall interest stated and broader angle of your findings, we are able to invite you to revise your manuscript experimentally to address the referees' comments. However, please note that the extent of revisions requested appear threshold in our view for the amount of complementary work we typically invite for our venue; also, I need to stress that we do require strong support from the referees on a revised version of the study in order to move on to publication of the work.

I would appreciate if you could contact me during the next weeks for exchange e.g. a video call to discuss your perspective on the comments and potential plan for revisions.

Please feel free to contact me if you have any questions or need further input on the referee comments.

When submitting your revised manuscript, please carefully review the instructions below.

Please feel free to approach me any time should you have additional questions related to this.

Thank you for the opportunity to consider your work for publication.

I look forward to your revision.

Kind regards,

Daniel Klimmeck

Daniel Klimmeck, PhD
Senior Editor
The EMBO Journal

Instruction for the preparation of your revised manuscript:

2) individual production quality figure files as .eps, .tif, .jpg (one file per figure).

3) a .docx formatted letter INCLUDING the reviewers' reports and your detailed point-by-point response to their comments. As part of the EMBO Press transparent editorial process, the point-by-point response is part of the Review Process File (RPF), which will be published alongside your paper.

4) a complete author checklist, which you can download from our author guidelines ([https://wol-prod-cdn.literatumonline.com/pb-assets/embo-site/Author Checklist%20-%20EMBO%20J-1561436015657.xlsx](https://wol-prod-cdn.literatumonline.com/pb-assets/embo-site/Author%20Checklist%20-%20EMBO%20J-1561436015657.xlsx)). Please insert information in the checklist that is also reflected in the manuscript. The completed author checklist will also be part of the RPF.

6) It is mandatory to include a 'Data Availability' section after the Materials and Methods. Before submitting your revision, primary datasets produced in this study need to be deposited in an appropriate public database, and the accession numbers and database listed under 'Data Availability'. Please remember to provide a reviewer password if the datasets are not yet public (see <https://www.embopress.org/page/journal/14602075/authorguide#datadeposition>).

7) Our journal encourages inclusion of *data citations in the reference list* to directly cite datasets that were re-used and obtained from public databases. Data citations in the article text are distinct from normal bibliographical citations and should directly link to the database records from which the data can be accessed. In the main text, data citations are formatted as follows: "Data ref: Smith et al, 2001" or "Data ref: NCBI Sequence Read Archive PRJNA342805, 2017". In the Reference list, data citations must be labeled with "[DATASET]". A data reference must provide the database name, accession number/identifiers and a resolvable link to the landing page from which the data can be accessed at the end of the reference. Further instructions are available at .

8) At EMBO Press we ask authors to provide source data for the main and EV figures. Our source data coordinator will contact you to discuss which figure panels we would need source data for and will also provide you with helpful tips on how to upload and organize the files.

Numerical data can be provided as individual .xls or .csv files (including a tab describing the data). For 'blots' or microscopy, uncropped images should be submitted (using a zip archive or a single pdf per main figure if multiple images need to be supplied for one panel). Additional information on source data and instruction on how to label the files are available at .

9) We replaced Supplementary Information with Expanded View (EV) Figures and Tables that are collapsible/expandable online (see examples in <https://www.embopress.org/doi/10.15252/emj.201695874>). A maximum of 5 EV Figures can be typeset. EV Figures should be cited as 'Figure EV1, Figure EV2' etc. in the text and their respective legends should be included in the main text after the legends of regular figures.

11) For data quantification: please specify the name of the statistical test used to generate error bars and P values, the number (n) of independent experiments (specify technical or biological replicates) underlying each data point and the test used to calculate p-values in each figure legend. The figure legends should contain a basic description of n, P and the test applied. Graphs must include a description of the bars and the error bars (s.d., s.e.m.).

The revision must be submitted online within 90 days; please click on the link below to submit the revision online before 10th Jul 2025.

Referee #1:

This manuscript presents an interesting and well-conducted study that explores the molecular mechanisms behind cachectic fat mass loss in the context of triple-negative breast cancer (TNBC). The authors demonstrate that secreted clusterin disrupts the circadian rhythm in white adipose tissue (WAT), leading to fat expenditure. This disruption is achieved through competitive binding of clusterin to 14-3-3, thereby inhibiting its interaction with the CLOCK/BMAL1 complex and sustaining its activation of circadian rhythm genes, particularly PER3. Furthermore, the manuscript suggests that the plakophilin 3 (PKP3)-clusterin axis stabilizes clusterin and promotes its secretion, thus contributing to cachexia in advanced TNBC. The potential therapeutic targeting of this axis to inhibit fat mass loss in preclinical models is a significant and novel aspect of the study.

I am unaware of studies exploring the function of clusterin in the context of cachexia and would thus deem the study as novel. The study offers important insights into the interplay between tumor-secreted factors and the disruption of circadian rhythms in adipose tissue during cancer cachexia. The experimental design appears solid, and the data presented support the hypothesis that the PKP3-clusterin axis plays a critical role in the pathophysiology of cancer-associated cachexia. The use of both in vitro and in vivo models adds robustness to the findings.

Overall, the manuscript is of high quality and presents novel findings that are of significant relevance to the field. However, I believe a few minor revisions would enhance clarity and provide further validation of the conclusions.

1. I am unable to find any information on how many animals have been used in the different experiments.
2. The manuscript lacks experiments that explore how the continuous activation of Per3 results in fat mass loss. In the discussion the authors discuss some relevant factors that might be driving the cachectic fat mass loss. It would be appreciated if the authors could measure them in their model (qPCR) to provide a link between circadian misalignment and the actual (patho)physiological response of the tissue. This angle is completely omitted in the current version of the manuscript.
3. The authors refer to "fat expenditure" throughout the manuscript to describe fat mass loss, which is a term that is entirely unfamiliar to me. Unless this is common practice in the cancer field, I would prefer "fat mass loss" or a similar expression to describe the fact that the adipose tissue shrinks/atrophies.
4. For readability please enlarge all H&E and tissue pictures throughout the manuscript.
5. The conclusions of this manuscript hinge on a large body of western blotting and Co-IPs. Please provide the catalog numbers of all your antibodies in the method section.
6. In Figures 1H & 4E & 6H, why can the authors not supply an absolute value for Clusterin levels in the adipose tissue?
7. Figure 2; injection studies. What circulating levels of Clu are achieved here?
8. Figure 2L: Can the adipocyte size distribution be determined using imageJ?
9. Many times throughout the manuscript sentences miss articles. For readability purposes I would recommend fixing those.
10. Figure 2E: The authors speculate that 14-3-3 prevents Bmal1/Clock proteins to enter into the nucleus which then dysregulates Per3, yet this western blot does not support this conclusion. Can the authors please comment on the discrepancy between Figure 2E and 2F?
11. The authors fail to discuss high-profile publications on the role of Clusterin in feeding behavior or diet-induced obesity, which contextualize its role in adipose tissue biology. <https://doi.org/10.1038/s41598-020-73927-y>,

Referee #2:

The manuscript by Liu and colleagues is an interesting follow up study to their prior work on CLU in TNB CA and there are interesting signals that tumor derived CLU may be affecting adipose tissue biology through tumor-fat communication. While there were several aspects to the study that were compelling, my more specific criticisms about this manuscript, which are listed below fall into three broad categories: (1) In my opinion, it does not provide a sufficient advance beyond their prior work to warrant publication in a high impact general audience journal; (2) The strength of the connection between CLU and cachexia independent of an antitumor effect is weak; (3) The adipose tissue mechanisms (which are arguably the central point of the paper) are not investigated in great depth.

- 1) The human data is presented as evidence that CLU is a marker and potential mechanistic driver of cachexia related to triple negative breast cancer. In this context, the cross-sectional comparison to healthy women does not address the question of cachexia related to cancer, but rather whether levels are elevated in cancer (which they have previously shown). If the authors want to address the question of cachexia, they would either need to have a breast cancer control group without cachexia, or do some sort of linear regression modeling based on quantifying the degree of cachexia (e.g. weight loss).
- 2) The fat phenotypes are generally under-developed with only one fat depot shown. It appears to be an inguinal depot based on appearance, but additional depots, including visceral and BAT should be shown.
- 3) Generally testing for circadian rhythms in gene expression should include at least 2 cycles - i.e. 48 hours.
- 4) Since the authors have previously shown that targeting the CLU pathway in tumors is a potential therapy, it is not clear that the effects of knocking the gene down in tumors is limiting fat contraction due to an on-target CLU effect in adipose versus simply an effective antitumor effect.
- 5) There is limited mechanistic depth to the adipose component of the story, as the primary readouts are fat mass and Per expression. How exactly is this affecting energy balance to cause weight loss? Is it an adipocyte autonomous effect (e.g. browning/beiging) or some other mechanism? Additional cell based mechanistic work, adipose tissue readouts, and whole body physiological studies (e.g. metabolic cages) would be required to start to understand mechanism, which would more definitively establish that tumor derived clusterin is a causal mediator of cachexia, rather than just a marker of severity of cancer. Based on what we know from adipose specific genetic disruption of circadian genes, it is not clear that PER over-expression in adipocytes should have such a dramatic effect on energy balance and adiposity.

Minor:

- 1) The authors have previously published human TNB ca data on CLU levels. Are the currently described human participants from the same prior study? If so, this needs to be stated clearly.
- 2) There is an over-use of dynamite plunger plots and lack of clarity on 'n' and statistical methods, which made it difficult to truly assess rigor.

Referee #3:

In "Tumor-secreted CLU induces cachectic fat expenditure via disrupting circadian rhythm in adipose", Liu and colleagues explore how clusterin (CLU) secreted by triple-negative breast cancer connects to white adipose tissue (WAT) and cachexia through the molecular circadian clock. They first note that tumor-secreted CLU is upregulated in cachectic TNBC models, and mediates loss of WAT. Interestingly, CLU seems to accomplish this by dysregulating CLOCK-BMAL1 binding to 14-3-3 and upregulating several circadian rhythm genes, including Per3. Elevated Per3 specifically regulated the loss of mass in WAT. CLU secretion in TNBC requires the accessory protein PKP3, and importantly, ectopic CLU can compensate for the loss of PKP3, putting it downstream of this accessory protein. Finally, the Authors show that this effect is present in human samples, and that cachexia can be reversed in TNBC PDXs with CLU or PKP3 antagonism. The Authors conclude that circadian disruption and Per3 elevation, mediated by CLU secretion, are partially responsible for cachexia and loss of WAT adiposity in TNBC. Overall, this is an outstanding study. The experiments are well-designed, clear, and backed up by replicates and proper controls. The mechanism of PKP3-CLU-Per3 in WAT is very interesting, and the human and PDX data at the end of the paper add immediate clinical relevance to the study. The work is of very high significance to the field and can make an immediate impact. My only major concern is an overinterpretation of the circadian rhythm connection. The Authors very nicely showed that CLU suppresses circadian gene expression, and proposed (but did not test) a specific mechanism through 14-3-3 binding of Clock and Bmal1. But the conclusions go far beyond the actual findings, and really the most solid finding is that elevated Per3 is connected to the phenotypes they see, and anything about circadian disruption needs to remain speculative. Because of the

very high significance of this work already, I explained how the Authors could make text changes to better focus on their actual results, but I also suggested experiments if they want to better prove that circadian disruption is connected to their phenotype. I also have a few minor comments I would like to see addressed. I look forward to reviewing a revised manuscript.

Major comments

The paper over-reaches with the Title and conclusions that circadian rhythm disruption is responsible for cachectic fat expenditure. Figure 2 very convincingly shows that the oscillation of Per3 is disrupted and it is elevated when CLU is present. However, the fact that Clock and Bmal1 still oscillate as normal in shCTRL very strongly argues AGAINST disruption of circadian rhythms as a system. Additionally, the Authors show that Clock and Bmal1 more strongly bind 14-3-3 when CLU is knocked down, but never prove that this is the mechanism that leads to increased Per3 expression (many other transcription factors probably regulate Per3). The results with Per3 are themselves strong enough for this paper, so I present two options to fix this overinterpretation:

1) Change the title and paper to reflect that the Authors show that elevated Per3 is responsible for changes in WAT, and have some evidence that circadian disruption may somehow be involved, but future studies are needed to dissect this (some of this can go in the Discussion).

2) Do more experiments to support the conclusion that circadian disruption is connected to WAT phenotype. Knock down 14-3-3, show that this increases Clock-Bmal1 nuclear localization + shCLU, and that other genes besides Per3 have disrupted rhythms in control conditions.

Minor comments

- Some of the "circadian rhythm genes" included in Figures 2B-C and 4F are not primarily molecular circadian clock genes, and should be excluded. The genes that should be excluded are: Usp2, Prox1, and Top2a. While these genes may regulate circadian rhythm, they do not belong on a list with the other genes like Dbp and Per3, which are core members of the circadian pathway. Excluding these genes will not change any of the conclusions of the paper.
- It is not clear why some genes in 2C are in red boxes.
- All IF and IHC images should be quantified, and statistical tests performed on the relevant comparisons.
- For Figure 4 and S2A, it would be helpful to show statistics on the individual comparisons. It is unclear if shCTRL is significantly different from shCLU and PKP3 only, or also from shPKP3 + Clu.
- For Figure 6, what is "anti-PKP3" and "anti-CLU"? The paper says these are lentiviruses. Are they shRNA, blocking antibodies, or something else? Please clarify in the text and Methods.

1st Editorial Decision**11th April 2025**

Dear Dr Liu,

Thank you again for the submission of your manuscript (EMBOJ-2025-120556) to The EMBO Journal, as well as for your patience with our feedback at this time. As mentioned earlier, your study was assessed by three reviewers with expertise in tumor biology, circadian control and adipogenesis, whose comments are enclosed below.

As you will see from the experts' reports, the referees acknowledge the analysis and potential interest and value of your findings. However, they also express major concerns regarding completeness and in vivo relevance of the findings, which need to be addressed thoroughly to make them supportive of publication in the EMBO Journal. In more detail, reviewer #2 raises substantial issues related to proof of a distinctive cachexia- vs broader tumor-regulatory role of CLU (ref#2, standfirst, pt.4); s/he also points to insufficient insights into the adipocyte biology affected by CLU (ref#2, pts. 2,5). Referee #3 agrees in that the mechanistic details provided need to be strengthened (ref#3, standfirst). Further, the reviewers raise a number of issues related to the human control data sets, presentation of the findings, improved methods annotation required, statistics applied and overall discussion of related literature, that would need to be conclusively addressed to achieve the level of robustness and clarity needed for The EMBO Journal.

Given the overall interest stated and broader angle of your findings, we are able to invite you to revise your manuscript experimentally to address the referees' comments. However, please note that the extent of revisions requested appear threshold in our view for the amount of complementary work we typically invite for our venue; also, I need to stress that we do require strong support from the referees on a revised version of the study in order to move on to publication of the work.

I would appreciate if you could contact me during the next weeks for exchange e.g. a video call to discuss your perspective on the comments and potential plan for revisions.

Please feel free to contact me if you have any questions or need further input on the referee comments.

We generally allow three months as standard revision time. As a matter of policy, competing manuscripts published during this period will not negatively impact on our assessment of the conceptual advance presented by your study. However, we request that you contact the editor as soon as possible upon publication of any related work, to discuss how to proceed. Should you foresee a problem in meeting this three-month deadline, please let us know in advance and we

may be able to grant an extension.

When submitting your revised manuscript, please carefully review the instructions below.

Please feel free to approach me any time should you have additional questions related to this.

Thank you for the opportunity to consider your work for publication.

I look forward to your revision.

Kind regards,

Daniel Klimmeck

Daniel Klimmeck, PhD
Senior Editor
The EMBO Journal

Instruction for the preparation of your revised manuscript:

2) individual production quality figure files as .eps, .tif, .jpg (one file per figure).

3) a .docx formatted letter INCLUDING the reviewers' reports and your detailed point-by-point response to their comments. As part of the EMBO Press transparent editorial process, the point-by-point response is part of the Review Process File (RPF), which will be published alongside your paper.

4) a complete author checklist, which you can download from our author guidelines ([https://wol-prod-cdn.literatumonline.com/pb-assets/embo-site/Author Checklist%20-%20EMBO%20J-1561436015657.xlsx](https://wol-prod-cdn.literatumonline.com/pb-assets/embo-site/Author%20Checklist%20-%20EMBO%20J-1561436015657.xlsx)). Please insert information in the checklist that is also reflected in the manuscript. The completed author checklist will also be part of the RPF.

6) It is mandatory to include a 'Data Availability' section after the Materials and Methods. Before submitting your revision, primary datasets produced in this study need to be deposited in an appropriate public database, and the accession numbers and database listed under 'Data Availability'. Please remember to provide a reviewer password if the datasets are not yet public (see <https://www.embopress.org/page/journal/14602075/authorguide#datadeposition>).

7) Our journal encourages inclusion of *data citations in the reference list* to directly cite datasets that were re-used and obtained from public databases. Data citations in the article text are distinct from normal bibliographical citations and should directly link to the database records from which the data can be accessed. In the main text, data citations are formatted as follows: "Data ref: Smith et al, 2001" or "Data ref: NCBI Sequence Read Archive PRJNA342805, 2017". In the Reference list, data citations must be labeled with "[DATASET]". A data reference must provide the database name, accession number/identifiers and a resolvable link to the landing page from which the data can be accessed at the end of the reference. Further instructions are available at <https://www.embopress.org/page/journal/14602075/authorguide#referencesformat> >.

8) At EMBO Press we ask authors to provide source data for the main and EV figures. Our source data coordinator will contact you to discuss which figure panels we would need source data for and will also provide you with helpful tips on how to upload and organize the files.

Numerical data can be provided as individual .xls or .csv files (including a tab describing the data). For 'blots' or microscopy, uncropped images should be submitted (using a zip archive or a single pdf per main figure if multiple images need to be supplied for one panel). Additional information on source data and instruction on how to label the files are available at <https://www.embopress.org/page/journal/14602075/authorguide#sourcedata> >.

9) We replaced Supplementary Information with Expanded View (EV) Figures and Tables that are collapsible/expandable online (see examples in <https://www.embopress.org/doi/10.15252/embj.201695874>). A maximum of 5 EV Figures can be typeset. EV Figures should be cited as 'Figure EV1, Figure EV2" etc. in the text and their respective legends should be included in the main text after the legends of regular figures.

- For the figures that you do NOT wish to display as Expanded View figures, they should be bundled together with their legends in a single PDF file called *Appendix*, which should start with a short Table of Content. Appendix figures should be referred to in the main text as: "Appendix Figure S1, Appendix Figure S2" etc. See detailed instructions regarding expanded view here: <https://www.embopress.org/page/journal/14602075/authorguide#expandedview> >.

11) For data quantification: please specify the name of the statistical test used to generate error bars and P values, the number (n) of independent experiments (specify technical or biological replicates) underlying each data point and the test used to calculate p-values in each figure legend. The figure legends should contain a basic description of n, P and the test applied. Graphs must include a description of the bars and the error bars (s.d., s.e.m.).

The revision must be submitted online within 90 days; please click on the link below to submit the revision online before 10th Jul 2025.

Referee #1:

This manuscript presents an interesting and well-conducted study that explores the molecular mechanisms behind cachectic fat mass loss in the context of triple-negative breast cancer (TNBC). The authors demonstrate that secreted clusterin disrupts the circadian rhythm in white adipose tissue (WAT), leading to fat expenditure. This disruption is achieved through competitive binding of clusterin to 14-3-3, thereby inhibiting its interaction with the CLOCK/BMAL1 complex and sustaining its activation of circadian rhythm genes, particularly PER3. Furthermore, the manuscript suggests that the plakophilin 3 (PKP3)-clusterin axis stabilizes clusterin and promotes its secretion, thus contributing to cachexia in advanced TNBC. The potential therapeutic targeting of this axis to inhibit fat mass loss in preclinical models is a significant and novel aspect of the study.

I am unaware of studies exploring the function of clusterin in the context of cachexia and would thus deem the study as novel. The study offers important insights into the interplay between tumor-secreted factors and the disruption of circadian rhythms in adipose tissue during cancer cachexia. The experimental design appears solid, and the data presented support the hypothesis that the PKP3-clusterin axis plays a critical role in the pathophysiology of cancer-associated cachexia. The use of both in vitro and in vivo models adds robustness to the findings.

Overall, the manuscript is of high quality and presents novel findings that are of significant relevance to the field. However, I believe a few minor revisions would enhance clarity and provide further validation of the conclusions.

1. I am unable to find any information on how many animals have been used in the different experiments.
2. The manuscript lacks experiments that explore how the continuous activation of Per3 results in fat mass loss. In the discussion the authors discuss some relevant factors that might be driving the cachectic fat mass loss. It would be appreciated if the authors could measure them in their model (qPCR) to provide a link between circadian misalignment and the actual (patho)physiological response of the tissue. This angle is completely omitted in the current version of the manuscript.
3. The authors refer to "fat expenditure" throughout the manuscript to describe fat mass loss, which is a term that is entirely unfamiliar to me. Unless this is common practice in the cancer field, I would prefer "fat mass loss" or a similar expression to describe the fact that the adipose tissue shrinks/atrophies.
4. For readability please enlarge all H&E and tissue pictures throughout the manuscript.
5. The conclusions of this manuscript hinge on a large body of western blotting and Co-IPs. Please provide the catalog numbers of all your antibodies in the method section.
6. In Figures 1H & 4E & 6H, why can the authors not supply an absolute value for Clusterin levels in the adipose tissue?
7. Figure 2; injection studies. What circulating levels of Clu are achieved here?
8. Figure 2L: Can the adipocyte size distribution be determined using imageJ?
9. Many times throughout the manuscript sentences miss articles. For readability purposes I would recommend fixing those.
10. Figure 2E: The authors speculate that 14-3-3 prevents Bmal1/Clock proteins to enter into the

nucleus which then dysregulates Per3, yet this western blot does not support this conclusion. Can the authors please comment on the discrepancy between Figure 2E and 2F?

11. The authors fail to discuss high-profile publications on the role of Clusterin in feeding behavior or diet-induced obesity, which contextualize its role in adipose tissue biology. <https://doi.org/10.1038/s41598-020-73927-y>, <https://doi.org/10.1038/ncomms2896>.

Referee #2:

The manuscript by Liu and colleagues is an interesting follow up study to their prior work on CLU in TNB CA and there are interesting signals that tumor derived CLU may be affecting adipose tissue biology through tumor-fat communication. While there were several aspects to the study that were compelling, my more specific criticisms about this manuscript, which are listed below fall into three broad categories: (1) In my opinion, it does not provide a sufficient advance beyond their prior work to warrant publication in a high impact general audience journal; (2) The strength of the connection between CLU and cachexia independent of an antitumor effect is weak; (3) The adipose tissue mechanisms (which are arguably the central point of the paper) are not investigated in great depth.

1) The human data is presented as evidence that CLU is a marker and potential mechanistic driver of cachexia related to triple negative breast cancer. In this context, the cross-sectional comparison to healthy women does not address the question of cachexia related to cancer, but rather whether levels are elevated in cancer (which they have previously shown). If the authors want to address the question of cachexia, they would either need to have a breast cancer control group without cachexia, or do some sort of linear regression modeling based on quantifying the degree of cachexia (e.g. weight loss).

2) The fat phenotypes are generally under-developed with only one fat depot shown. It appears to be an inguinal depot based on appearance, but additional depots, including visceral and BAT should be shown.

3) Generally testing for circadian rhythms in gene expression should include at least 2 cycles - i.e. 48 hours.

4) Since the authors have previously shown that targeting the CLU pathway in tumors is a potential therapy, it is not clear that the effects of knocking the gene down in tumors is limiting fat contraction due to an on-target CLU effect in adipose versus simply an effective antitumor effect.

5) There is limited mechanistic depth to the adipose component of the story, as the primary readouts are fat mass and Per expression. How exactly is this affecting energy balance to cause weight loss? Is it an adipocyte autonomous effect (e.g. browning/beiging) or some other mechanism? Additional cell based mechanistic work, adipose tissue readouts, and whole body physiological studies (e.g. metabolic cages) would be required to start to understand mechanism,

which would more definitively establish that tumor derived clusterin is a causal mediator of cachexia, rather than just a marker of severity of cancer. Based on what we know from adipose specific genetic disruption of circadian genes, it is not clear that PER over-expression in adipocytes should have such a dramatic effect on energy balance and adiposity.

Minor:

1) The authors have previously published human TNB ca data on CLU levels. Are the currently described human participants from the same prior study? If so, this needs to be stated clearly.

2) There is an over-use of dynamite plunger plots and lack of clarity on 'n' and statistical methods, which made it difficult to truly assess rigor.

Referee #3:

In "Tumor-secreted CLU induces cachectic fat expenditure via disrupting circadian rhythm in adipose", Liu and colleagues explore how clusterin (CLU) secreted by triple-negative breast cancer connects to white adipose tissue (WAT) and cachexia through the molecular circadian clock. They first note that tumor-secreted CLU is upregulated in cachectic TNBC models, and mediates loss of WAT. Interestingly, CLU seems to accomplish this by dysregulating CLOCK-BMAL1 binding to 14-3-3 and upregulating several circadian rhythm genes, including Per3. Elevated Per3 specifically regulated the loss of mass in WAT. CLU secretion in TNBC requires the accessory protein PKP3, and importantly, ectopic CLU can compensate for the loss of PKP3, putting it downstream of this accessory protein. Finally, the Authors show that this effect is present in human samples, and that cachexia can be reversed in TNBC PDXs with CLU or PKP3 antagonism. The Authors conclude that circadian disruption and Per3 elevation, mediated by CLU secretion, are partially responsible for cachexia and loss of WAT adiposity in TNBC.

Overall, this is an outstanding study. The experiments are well-designed, clear, and backed up by replicates and proper controls. The mechanism of PKP3-CLU-Per3 in WAT is very interesting, and the human and PDX data at the end of the paper add immediate clinical relevance to the study. The work is of very high significance to the field and can make an immediate impact. My only major concern is an overinterpretation of the circadian rhythm connection. The Authors very nicely showed that CLU suppresses circadian gene expression, and proposed (but did not test) a specific mechanism through 14-3-3 binding of Clock and Bmal1. But the conclusions go far beyond the actual findings, and really the most solid finding is that elevated Per3 is connected to the phenotypes they see, and anything about circadian disruption needs to remain speculative. Because of the very high significance of this work already, I explained how the Authors could make text changes to better focus on their actual results, but I also suggested experiments if they want to better prove that circadian disruption is connected to their phenotype. I also have a few minor comments I would like to see addressed. I look forward to reviewing a revised manuscript.

Major comments

The paper over-reaches with the Title and conclusions that circadian rhythm disruption is

responsible for cachectic fat expenditure. Figure 2 very convincingly shows that the oscillation of Per3 is disrupted and it is elevated when CLU is present. However, the fact that Clock and Bmal1 still oscillate as normal in shCTRL very strongly argues AGAINST disruption of circadian rhythms as a system. Additionally, the Authors show that Clock and Bmal1 more strongly bind 14-3-3 when CLU is knocked down, but never prove that this is the mechanism that leads to increased Per3 expression (many other transcription factors probably regulate Per3). The results with Per3 are themselves strong enough for this paper, so I present two options to fix this overinterpretation:

1) Change the title and paper to reflect that the Authors show that elevated Per3 is responsible for changes in WAT, and have some evidence that circadian disruption may somehow be involved, but future studies are needed to dissect this (some of this can go in the Discussion).

2) Do more experiments to support the conclusion that circadian disruption is connected to WAT phenotype. Knock down 14-3-3, show that this increases Clock-Bmal1 nuclear localization + shCLU, and that other genes besides Per3 have disrupted rhythms in control conditions.

Minor comments

- Some of the "circadian rhythm genes" included in Figures 2B-C and 4F are not primarily molecular circadian clock genes, and should be excluded. The genes that should be excluded are: Usp2, Prox1, and Top2a. While these genes may regulate circadian rhythm, they do not belong on a list with the other genes like Dbp and Per3, which are core members of the circadian pathway. Excluding these genes will not change any of the conclusions of the paper.
- It is not clear why some genes in 2C are in red boxes.
- All IF and IHC images should be quantified, and statistical tests performed on the relevant comparisons.
- For Figure 4 and S2A, it would be helpful to show statistics on the individual comparisons. It is unclear if shCTRL is significantly different from shCLU and PKP3 only, or also from shPKP3 + Clu.
- For Figure 6, what is "anti-PKP3" and "anti-CLU"? The paper says these are lentiviruses. Are they shRNA, blocking antibodies, or something else? Please clarify in the text and Methods.

1st Authors' Response to Reviewers

Response to Reviewers' comments

Referee #1:

This manuscript presents an interesting and well-conducted study that explores the molecular mechanisms behind cachectic fat mass loss in the context of triple-negative breast cancer (TNBC). The authors demonstrate that secreted clusterin disrupts the circadian rhythm in white adipose tissue (WAT), leading to fat expenditure. This disruption is achieved through competitive binding of clusterin to 14-3-3, thereby inhibiting its interaction with the CLOCK/BMAL1 complex and sustaining its activation of circadian rhythm genes, particularly PER3. Furthermore, the manuscript suggests that the plakophilin 3 (PKP3)-clusterin axis stabilizes clusterin and promotes its secretion, thus contributing to cachexia in advanced TNBC. The potential therapeutic targeting of this axis to inhibit fat mass loss in preclinical models is a significant and novel aspect of the study.

I am unaware of studies exploring the function of clusterin in the context of cachexia and would thus deem the study as novel. The study offers important insights into the interplay between tumor-secreted factors and the disruption of circadian rhythms in adipose tissue during cancer cachexia. The experimental design appears solid, and the data presented support the hypothesis that the PKP3-clusterin axis plays a critical role in the pathophysiology of cancer-associated cachexia. The use of both in vitro and in vivo models adds robustness to the findings.

Overall, the manuscript is of high quality and presents novel findings that are of significant relevance to the field. However, I believe a few minor revisions would enhance clarity and provide further validation of the conclusions.

We are grateful to the reviewer for the strong recommendation of minor revision with recognizing the significance and interest of our study and other strengths. Our response to the specific comments and suggestions raised by the Reviewer is as follows:

1. I am unable to find any information on how many animals have been used in the different experiments.

Response: We appreciate the Reviewer's comments. Each group contains at least five mice. The number of animals used in different experiments has been provided in the figure legends, including the figure legends of Figures 1, 2, 4, 6 and EV5 in our revised manuscript.

2. The manuscript lacks experiments that explore how the continuous activation of Per3 results in fat mass loss. In the discussion the authors discuss a some relevant

factors that might be driving the cachectic fat mass loss. It would be appreciated if the authors could measure them in their model (qPCR) to provide a link between circadian misalignment and the actual (patho)physiological response of the tissue. This angle is completely omitted in the current version of the manuscript.

Response: We appreciate the Reviewer's constructive comments with experimental suggestions. As suggested, we have performed RT-qPCR to detect genes related to fat mass loss in the iWAT of mice. The results showed that the expression levels of adipogenesis-related genes were increased in the iWAT of mice bearing CLU knockdown tumors compared to controls (Figure EV3A). In support, mice with intravenous injection of secreted CLU purified protein showed significantly decreased expression levels of adipogenesis-related genes in iWAT compared to the control, and additional knockdown of Per3 in iWAT alleviated the secreted CLU-decreased expression levels of adipogenesis-related genes (Figure EV3B). We have added these new data with clear description in our revised manuscript.

Figure EV3.....(A) Adipogenesis-related gene expression by RT-qPCR in iWAT of MDA-MB-468 tumor-bearing mice with indicated treatment (shCTRL vs shCLU: PPARG, $P=0.0003$; Adifoq, $P<0.0001$; Plin1, $P<0.0001$; Fabp4, $P=0.0002$; AP2, $P<0.0001$; PPAR γ 2, $P<0.0001$); **(B)** Adipogenesis-related gene expression by RT-qPCR in iWAT of healthy nude mice with indicated treatment (CTRL vs CLU: PPARG, $P=0.0002$; Adifoq, $P<0.0001$; Plin1, $P=0.0002$; Fabp4, $P=0.0001$; AP2, $P=0.0002$; PPAR γ 2, $P=0.0003$; CTRL vs CLU+siPer3: PPARG, $P=0.0142$; Adifoq, $P=0.0018$; Plin1, $P=0.1590$; Fabp4, $P=0.2300$; AP2, $P=0.0088$; PPAR γ 2, $P=0.0163$).

3. The authors refer to "fat expenditure" throughout the manuscript to describe fat mass loss, which is a term that is entirely unfamiliar to me. Unless this is common practice in the cancer field, I would prefer "fat mass loss" or a similar expression to describe the fact that the adipose tissue shrinks/atrophies.

Response: Thank you for your comments and suggestions. As suggested, we have used "fat mass loss" to replace "fat expenditure" throughout our revised manuscript.

4. For readability please enlarge all H&E and tissue pictures throughout the manuscript.

Response: Thank you for your suggestions. As requested, we have enlarged all H&E

and tissue pictures throughout our revised manuscript.

5. The conclusions of this manuscript hinge on a large body of western blotting and Co-IPs. Please provide the catalog numbers of all your antibodies in the method section.

Response: We thank for the reviewer's comments and suggestions. As requested, we have added the catalog numbers of all antibodies used in Western blotting and Co-IPs in the method section in our revised manuscript.

6. In Figures 1H & 4E & 6H, why can the authors not supply an absolute value for Clusterin levels in the adipose tissue?

Response: We thank for the reviewer's comments and suggestions. As suggested, we have supplied absolute value for clusterin levels in the adipose tissue in Figures 1H & 4E & 6H in our revised manuscript.

7. Figure 2; injection studies. What circulating levels of Clu are achieved here?

Response: Thank you for your comments. We achieved circulation levels of Clu for injection studies in Figure 2 via based on the serum CLU level in TNBC patients with cachexia and tumor bearing mice with cachexia, and Western blotting assay showed that circulating CLU levels in iWAT in injection studies were comparable to those in iWAT of tumor bearing mice with cachexia (Figure EV2D). We have added these new data with a clear description in our revised manuscript.

Figure EV2. ...*(D)* circulating CLU levels in iWAT of mice by Western blot.

8. Figure 2L: Can the adipocyte size distribution be determined using imageJ?

Response: Thank you for your comments and suggestions. As suggested, we have used imageJ to determine the adipocyte size distribution in Figure 2L and in other figures, including Figures 1G, 4D and 6G. We have added these new data with a clear description in our revised manuscript (Figures EV1, EV2E, EV5D, and EV5J in our revised manuscript).

Figure EV1. Statistical analysis of H&E staining of iWAT, gWAT, BAT in Figure 1G (shCTRL vs shCLU: iWAT, $P<0.0001$; gWAT, $P=0.0014$; BAT, $P=0.0229$). The data are presented as mean \pm SEM. * $p<0.05$, ** $p<0.01$, *** $p<0.001$. Statistical comparisons between groups were performed using unpaired two-tailed Student's t tests. Source data are available online for this figure.

Figure EV2. ...*(E)* Statistical analysis of H&E staining of iWAT, gWAT, BAT in Figure 2L (iWAT: CTRL vs CLU, $P<0.0001$; CTRL vs CLU+siPer3, $P=0.0016$; CLU vs CLU+siPer3, $P=0.0151$; gWAT: CTRL vs CLU, $P<0.0001$; CTRL vs CLU+siPer3, $P=0.0057$; CLU vs CLU+siPer3, $P=0.0819$; BAT: CTRL vs CLU, $P=0.0137$; CTRL vs CLU+siPer3, $P=0.0511$; CLU vs CLU+siPer3, $P=0.5122$). The data are presented as mean \pm SEM. * $p<0.05$, ** $p<0.01$, *** $p<0.001$, ns for not significant. Statistical comparisons between groups were performed using unpaired two-tailed Student's t tests. Source data are available online for this figure.

Figure EV5. ...*(D)* Statistical analysis of H&E staining of iWAT, gWAT, BAT in Figure 4D (iWAT: shCTRL vs shPKP3, $P<0.0001$; shCTRL vs shCLU, $P<0.0001$; shCTRL vs shPKP3+CLU, $P=0.7042$; gWAT: shCTRL vs shPKP3, $P<0.0001$; shCTRL vs shCLU, $P<0.0001$; shCTRL vs shPKP3+CLU, $P=0.0009$; BAT: shCTRL vs shPKP3, $P=0.2816$; shCTRL vs shCLU, $P=0.1173$; shCTRL vs shPKP3+CLU, $P=0.0980$).

Figure EV5...(J) Statistical analysis of H&E staining of iWAT, gWAT, BAT in Figure 6G (iWAT: shCTRL vs shPKP3, $P < 0.0001$; shCTRL vs shCLU, $P < 0.0001$; gWAT: shCTRL vs shPKP3, $P = 0.0002$; shCTRL vs shCLU, $P = 0.0068$; BAT: shCTRL vs shPKP3, $P = 0.0745$; shCTRL vs shCLU, $P = 0.1963$). The data are presented as mean \pm SEM. ** $p < 0.01$, *** $p < 0.001$, ns for not significant. Statistical comparisons between groups were performed using unpaired two-tailed Student's t tests.

9. Many times throughout the manuscript sentences miss articles. For readability purposes I would recommend fixing those.

Response: Thank you for your careful readings. As recommended, we have corrected these sentences throughout the manuscript in our revised manuscript.

10. Figure 2E: The authors speculate that 14-3-3 prevents Bmal1/Clock proteins to enter into the nucleus which then dysregulates Per3, yet this western blot does not support this conclusion. Can the authors please comment on the discrepancy between Figure 2E and 2F?

Response: We thank for the reviewer's comments. We speculated that the discrepancy between Figures 2E and 2F is due in part to the cytoplasm-to-nucleus translocation behavior of Bmal1/Clock. To address this, we performed nucleocytoplasmic separation assay in the iWAT derived from mice bearing CLU knockdown tumors. The results showed that protein level of Clock/Bmal1 increased in the nucleus of iWAT cells with 14-3-3 knockdown compared to iWAT without 14-3-3 knockdown (Figure EV2B). Furthermore, the expression level of Per3 was restored in the iWAT with 14-3-3 knockdown compared to iWAT without 14-3-3 knockdown (Figure EV2C). These results support our speculation that 14-3-3 is important for preventing Bmal1/Clock protein from entering into the nucleus for the *Per3* dysregulation. We have added these new data with clear description in our revised manuscript.

B

Figure EV2. ...(B) Representative Western blot images of a nucleocytoplasmic separation assay showing the levels of cytoplasmic and nuclear Clock, Bmal1, and 14-3-3 in iWAT cells with and without 14-3-3 knockdown in mice bearing CLU knockdown tumors.

Figure EV2. ...(C) Per3 expression in iWAT cells with and without 14-3-3 knockdown by RT-qPCR. The shCLU and shCTRL stand mice bearing tumor with and without CLU knockdown, respectively (shCTRL+siCTRL vs shCTRL+si14-3-3: Per3, $P=0.0691$; shCLU+siCTRL vs shCLU+si14-3-3: Per3, $P=0.0001$).

11. The authors fail to discuss high-profile publications on the role of Clusterin in feeding behavior or diet-induced obesity, which contextualize its role in adipose tissue

biology. <https://doi.org/10.1038/s41598-020-73927-y>, <https://doi.org/10.1038/ncomms2896>.

Response: Thank you for your comments and suggestions. We have cited and discussed these high-profile publications on the role of Clusterin in feeding behavior or diet-induced obesity in our revised manuscript.

Referee #2:

The manuscript by Liu and colleagues is an interesting follow up study to their prior work on CLU in TNB CA and there are interesting signals that tumor derived CLU may be affecting adipose tissue biology through tumor-fat communication. While there were several aspects to the study that were compelling, my more specific criticisms about this manuscript, which are listed below fall into three broad categories: (1) In my opinion, it does not provide a sufficient advance beyond their prior work to warrant publication in a high impact general audience journal; (2) The strength of the connection between CLU and cachexia independent of an antitumor effect is weak; (3) The adipose tissue mechanisms (which are arguably the central point of the paper) are not investigated in great depth.

Thanks for the reviewer's recognizing the several compelling aspects of our study and valuable specific criticisms. To address these points objectively and scientifically, we

have performed all the experiments suggested by the reviewer.

1) The human data is presented as evidence that CLU is a marker and potential mechanistic driver of cachexia related to triple negative breast cancer. In this context, the cross-sectional comparison to healthy women does not address the question of cachexia related to cancer, but rather whether levels are elevated in cancer (which they have previously shown). If the authors want to address the question of cachexia, they would either need to have a breast cancer control group without cachexia, or do some sort of linear regression modeling based on quantifying the degree of cachexia (e.g. weight loss).

Response: We thank for the reviewer's comments with experimental suggestions. As suggested, we have investigated CLU expression in the serum of TNBC patient with or without cachexia. The results showed that the level of secreted CLU in the serum of TNBC patients with cachexia is significantly higher than that of TNBC patient without cachexia (Figure 1B). We have updated these new data with clear description in our revised manuscript.

Figure 1. ...*(B)* Secreted CLU levels in serum samples from cachectic TNBC patients compared to healthy women, slim healthy women, and non-cachectic TNBC patients by ELISA assay (CTRL, $n=22$; Slim CTRL, $n=21$; Non-cachectic TNBC, $n=21$; Cachectic TNBC, $n=20$; CTRL vs Non-cachectic TNBC, $P=0.0009$; CTRL vs Cachectic TNBC, $P<0.0001$; Slim CTRL vs Non-cachectic TNBC, $P=0.0007$; Slim CTRL vs Cachectic TNBC, $P<0.0001$; Non-cachectic TNBC vs Cachectic TNBC, $P=0.0055$).

2) The fat phenotypes are generally under-developed with only one fat depot shown. It appears to be an inguinal depot based on appearance, but additional depots, including visceral and BAT should be shown.

Response: Thank you for your comments and suggestions. As requested, we have provided the additional depots, including visceral and BAT in our revised figures, including Figures 1F, 2K, 4C, 6F and EV5C in our revised manuscript.

Figure 1. ...*(F)* Representative iWAT, gWAT, BAT photos, Scale bar: 1 cm;

Figure 2. ...*(K)* Representative iWAT, gWAT, BAT photos, Scale bar: 1 cm;

Figure 4. ...*(C)* Representative iWAT, gWAT, BAT photos, Scale bar: 1 cm;

Figure 6. ...*(F)* Representative iWAT, gWAT, BAT photos, Scale bar: 1 cm;

Figure EV5. ...**(C)** Representative iWAT, gWAT, BAT photos, Scale bar: 1 cm;

3) Generally testing for circadian rhythms in gene expression should include at least 2 cycles - i.e. 48 hours.

Response: We thank for the reviewer's comments and suggestions. As requested, we have provided the gene expression of 2 cycles (i.e. 48 hours) in our revised figures, including Figures 2D-E, 2H-I, 4G-J, 6I-J, and EV5F-I in our revised manuscript.

Figure 2. ...**(D-E)** CLOCK, BMAL1, PER3 expression in a 48-hour cycle by **(D)** RT-qPCR and **(E)** Western blot assay;

Figure 2. ...**(H-I)** PER3 expression in a 48-hour cycle by **(H)** RT-qPCR and **(I)** Western blot assay;

Figure 4. ...**(G-J)** CLOCK, BMAL1, PER3 expression in iWAT in a 48-hour cycle by **(G-I)** RT-qPCR and **(J)** Western blot assay.

Figure 6. ...**(I-J)** CLOCK, BMAL1, PER3 expression in a 48-hour cycle by **(I)** RT-qPCR and **(J)** Western blot assay.

Figure EV5. ...**(F-I)** CLOCK, BMAL1, PER3 expression in WAT in a 48-hour cycle by **(F-H)** RT-qPCR and **(I)** Western blot assay.

4) Since the authors have previously shown that targeting the CLU pathway in tumors is a potential therapy, it is not clear that the effects of knocking the gene down in tumors is limiting fat contraction due to an on-target CLU effect in adipose versus simply an effective antitumor effect.

Response: Thank you for your comments. In addition to knocking the CLU gene down in tumors of tumor-bearing mice, to confirm secreted CLU promoted fat expenditure, in this work we have treated mice without tumors by intravenous injection of purified CLU protein. The results showed that intravenous injection of CLU purified protein significantly decreased both the weight and size of WAT compared to the control, and additional knockdown of Per3 in WAT alleviated the CLU-promoted fat expenditure (Fig. 2 J-L). Taken together, these results suggest that the high level of secreted CLU underlies fat contraction, and the reduction of tumoral CLU plays an important role in limiting fat contraction. We hope these data address your concerns. Thank you once again for your valuable feedback.

5) There is limited mechanistic depth to the adipose component of the story, as the primary readouts are fat mass and Per expression. How exactly is this affecting energy balance to cause weight loss? Is it an adipocyte autonomous effect (e.g. browning/beiging) or some other mechanism? Additional cell based mechanistic work, adipose tissue readouts, and whole body physiological studies (e.g. metabolic cages) would be required to start to understand mechanism, which would more definitively establish that tumor derived clusterin is a causal mediator of cachexia, rather than just

a marker of severity of cancer. Based on what we know from adipose specific genetic disruption of circadian genes, it is not clear that PER over-expression in adipocytes should have such a dramatic effect on energy balance and adiposity.

Response: We thank for the reviewer's comments with experimental suggestions. This comment is relevance to the *Comment 2 from reviewer#1*. As suggested, we have performed RT-qPCR to detect fat mass loss related genes in the iWAT of mice. The results showed increased expression levels of selected adipogenesis-related genes were increased in the iWAT of mice with CLU-knockdown tumors compared to controls (Figure EV3A). In comparison, the intravenous injection of secreted CLU purified protein significantly decreased adipogenesis gene expression compared to the control, where additional knockdown of Per3 in iWAT restored adipogenesis gene expression levels (Figure EV3B). Oil Red O staining assays showed consistent results (Figure EV3C-D). In addition to adipogenesis dysregulation, fat mass loss can be promoted by other mechanisms, such as increased energy expenditure. Using a metabolic cage assay, we explored energy expenditure and found decreased oxygen consumption, carbon dioxide production, and heat generation in mice with CLU-knockdown tumors compared to controls (Figure EV3E-G). Intravenous injection of secreted CLU purified protein significantly increased the oxygen consumption, carbon dioxide production, and heat generation of mice compared to the control, and additional knockdown of Per3 in iWAT alleviated the energy expenditure (Figure EV3H-J). These results suggest that both adipogenesis and energy expenditure contribute to the fat mass loss promoted by the CLU-Per3 axis. We have added these new data with clear description in our revised manuscript.

Figure EV3. The CLU-Per3 axis contribute to adipogenesis and energy expenditure. (A) Adipogenesis-related gene expression by RT-qPCR in iWAT of MDA-MB-468 tumor-bearing mice with indicated treatment (shCTRL vs shCLU: PPAR γ , $P=0.0003$; Adipoq, $P<0.0001$; Plin1, $P<0.0001$; Fabp4, $P=0.0002$; AP2, $P<0.0001$; PPAR γ 2, $P<0.0001$). (B) Adipogenesis-related gene expression by RT-qPCR in iWAT of healthy nude mice with indicated treatment (CTRL vs CLU: PPAR γ , $P=0.0002$; Adipoq, $P<0.0001$; Plin1, $P=0.0002$; Fabp4, $P=0.0001$; AP2, $P=0.0002$; PPAR γ 2, $P=0.0003$; CTRL vs CLU+siPer3: PPAR γ , $P=0.0142$; Adipoq, $P=0.0018$; Plin1, $P=0.1590$; Fabp4, $P=0.2300$; AP2, $P=0.0088$; PPAR γ 2, $P=0.0163$). (C-D) Adipogenesis by using Oil Red O staining in iWAT of mice. (E, H) Oxygen consumption, (F, I) carbon dioxide generation and (G, J) heat production by metabolic cages in mice. The data are presented as mean \pm SEM. * $p<0.05$, ** $p<0.01$, *** $p<0.001$, ns for not significant. Statistical comparisons between groups were performed using unpaired two-tailed Student's t tests. Source data are available online for this figure.

We hope these new data address your concerns. Thank you once again for your valuable comments and experimental suggestions.

Minor:

1) The authors have previously published human TNB ca data on CLU levels. Are the currently described human participants from the same prior study? If so, this needs to be stated clearly.

Response: We thank for the reviewer's comments and suggestions. The currently described human participants are a new set of patients, who have no overlap with those used in our prior study.

2) There is an over-use of dynamite plunger plots and lack of clarity on 'n' and statistical methods, which made it difficult to truly assess rigor.

Response: We thank for the reviewer's suggestions. As suggested, we have reduced the use of dynamite plunger plots and clarify "n" and statistical method in figure legend and methods section in our revised manuscript.

Referee #3:

In "Tumor-secreted CLU induces cachectic fat expenditure via disrupting circadian rhythm in adipose", Liu and colleagues explore how clusterin (CLU) secreted by triple-negative breast cancer connects to white adipose tissue (WAT) and cachexia through the molecular circadian clock. They first note that tumor-secreted CLU is upregulated in cachectic TNBC models, and mediates loss of WAT. Interestingly, CLU seems to accomplish this by dysregulating CLOCK-BMAL1 binding to 14-3-3 and upregulating several circadian rhythm genes, including Per3. Elevated Per3 specifically regulated the loss of mass in WAT. CLU secretion in TNBC requires the accessory protein PKP3, and importantly, ectopic CLU can compensate for the loss of PKP3, putting it downstream of this accessory protein. Finally, the Authors show that this effect is present in human samples, and that cachexia can be reversed in TNBC PDXs with CLU or PKP3 antagonism. The Authors conclude that circadian disruption and Per3 elevation, mediated by CLU secretion, are partially responsible for cachexia and loss of WAT adiposity in TNBC.

Overall, this is an outstanding study. The experiments are well-designed, clear, and backed up by replicates and proper controls. The mechanism of PKP3-CLU-Per3 in WAT is very interesting, and the human and PDX data at the end of the paper add immediate clinical relevance to the study. The work is of very high significance to the field and can make an immediate impact. My only major concern is an overinterpretation of the circadian rhythm connection. The Authors very nicely showed that CLU suppresses circadian gene expression, and proposed (but did not test) a specific mechanism through 14-3-3 binding of Clock and Bmal1. But the conclusions go far beyond the actual findings, and really the most solid finding is that elevated Per3 is connected to the phenotypes they see, and anything about circadian disruption needs to remain speculative. Because of the very high significance of this work already, I explained how the Authors could make text changes to better focus on their actual results, but I also suggested experiments if they want to better prove that circadian disruption is connected to their phenotype. I also have a few minor comments I would like to see addressed. I look forward to reviewing a revised manuscript.

We thank the Reviewer for the overall positive comments and additional valuable suggestions on our manuscript. Our response to their specific points is as follows:

Major comments

The paper over-reaches with the Title and conclusions that circadian rhythm disruption is responsible for cachectic fat expenditure. Figure 2 very convincingly shows that the oscillation of Per3 is disrupted and it is elevated when CLU is present. However, the fact that Clock and Bmal1 still oscillate as normal in shCTRL very strongly argues AGAINST disruption of circadian rhythms as a system. Additionally, the Authors show that Clock and Bmal1 more strongly bind 14-3-3 when CLU is knocked down, but never prove that this is the mechanism that leads to increased Per3 expression (many other transcription factors probably regulate Per3). The results with Per3 are themselves strong enough for this paper, so I present two options to fix this overinterpretation:

- 1) Change the title and paper to reflect that the Authors show that elevated Per3 is responsible for changes in WAT, and have some evidence that circadian disruption may somehow be involved, but future studies are needed to dissect this (some of this can go in the Discussion).
- 2) Do more experiments to support the conclusion that circadian disruption is connected to WAT phenotype. Knock down 14-3-3, show that this increases Clock-Bmal1 nuclear localization + shCLU, and that other genes besides Per3 have disrupted rhythms in control conditions.

Response: We thank for the reviewer's comments with providing two options to fix the potentially overinterpretation of our results. We selected the option 2) and do more experiments suggested by the reviewer to support the conclusion. The results showed that knocking down 14-3-3, show that this increases Clock-Bmal1 nuclear localization in the iWAT cells from shCLU mice (Figure EV2B). Additionally, other genes besides *Per3*, such as *Per 2*, showed a similar expression pattern to Per 3 (Figure for review only). We have added these new data with clear description in our revised manuscript.

B

Figure EV2. ...*(B)* Representative Western blot images of CLOCK, BMAL1 and 14-3-3 of mice bearing CLU knockdown tumors by nucleocytoplasmic separation

assays.

Figure for review only. *Per2* also showed disrupted rhythms in the iWAT of mice bearing tumor in control condition (shCTRL) compared to mice bearing CLU-knockdown tumor (shCLU). (A) *Per2* expression in iWAT of mice with indicated treatment by RT-qPCR (shCTRL+siCTRL vs shCTRL+si14-3-3: *Per2*, $P=0.0659$; shCLU+siCTRL vs shCLU+si14-3-3: *Per2*, $P=0.0017$). (B) *Per2* expression in a 24-hour cycle by RT-qPCR.

Minor comments

- Some of the "circadian rhythm genes" included in Figures 2B-C and 4F are not primarily molecular circadian clock genes, and should be excluded. The genes that should be excluded are: *Usp2*, *Prox1*, and *Top2a*. While these genes may regulate circadian rhythm, they do not belong on a list with the other genes like *Dbp* and *Per3*, which are core members of the circadian pathway. Excluding these genes will not change any of the conclusions of the paper.

Response: Thanks for the suggestion. As requested, we have removed these genes of Figure 2B-C and Figure 4F in our revised manuscript.

- It is not clear why some genes in 2C are in red boxes.

Response: Thanks for the suggestion. As suggested, we have removed red boxes in our revised manuscript.

- All IF and IHC images should be quantified, and statistical tests performed on the relevant comparisons.

Response: Thanks for the suggestion. As requested, we have quantified, and performed statistical tests on the relevant comparisons of all IF and IHC images. These results have been included in our revised manuscript (Figures EV1, EV2E, EV4M-T, EV5D and EV5J in our revised manuscript).

Figure EV1. Statistical analysis of H&E staining of iWAT, gWAT, BAT in Figure 1G (shCTRL vs shCLU: iWAT, $P < 0.0001$; gWAT, $P = 0.0014$; BAT, $P = 0.0229$). The data are presented as mean \pm SEM. * $p < 0.05$, ** $p < 0.01$, *** $p < 0.001$. Statistical comparisons between groups were performed using unpaired two-tailed Student's *t* tests. Source data are available online for this figure.

Figure EV2. ...*(E)* Statistical analysis of H&E staining of iWAT, gWAT, BAT in Figure 2M (iWAT: CTRL vs CLU, $P < 0.0001$; CTRL vs CLU+siPer3, $P = 0.0016$; CLU vs CLU+siPer3, $P = 0.0151$; gWAT: CTRL vs CLU, $P < 0.0001$; CTRL vs CLU+siPer3, $P = 0.0057$; CLU vs CLU+siPer3, $P = 0.0819$; BAT: CTRL vs CLU, $P = 0.0137$; CTRL vs CLU+siPer3, $P = 0.0511$; CLU vs CLU+siPer3, $P = 0.5122$). The data are presented as mean \pm SEM. * $p < 0.05$, ** $p < 0.01$, *** $p < 0.001$, ns for not significant. Statistical comparisons between groups were performed using unpaired two-tailed Student's *t* tests. Source data are available online for this figure.

Figure EV4. ...*(M)* Colocalization analysis of Figure 3F. *(O)* colocalization analysis showing PKP3 and DSG localization in MDA-MB-231 cells after PKP3 knockdown compared to the control. *(P)* Colocalization analysis of Figure 3G. *(R)* colocalization analysis showing PKP3 and CLU localization in MDA-MB-231 cells. *(S)*

Colocalization analysis of Figure 3H. (T) Colocalization analysis showing CLU and LAMP1 localization in MDA-MB-231 cells upon PKP3 knockdown compared to the control.

Figure EV5. ...*(D)* Statistical analysis of H&E staining of iWAT, gWAT, BAT in Figure 4D (iWAT: shCTRL vs shPKP3, $P<0.0001$; shCTRL vs shCLU, $P<0.0001$; shCTRL vs shPKP3+CLU, $P=0.7042$; gWAT: shCTRL vs shPKP3, $P<0.0001$; shCTRL vs shCLU, $P<0.0001$; shCTRL vs shPKP3+CLU, $P=0.0009$; BAT: shCTRL vs shPKP3, $P=0.2816$; shCTRL vs shCLU, $P=0.1173$; shCTRL vs shPKP3+CLU, $P=0.0980$).

Figure EV5. ...*(J)* Statistical analysis of H&E staining of iWAT, gWAT, BAT in Figure 6G (iWAT: shCTRL vs shPKP3, $P<0.0001$; shCTRL vs shCLU, $P<0.0001$; gWAT: shCTRL vs shPKP3, $P=0.0002$; shCTRL vs shCLU, $P=0.0068$; BAT: shCTRL vs shPKP3, $P=0.0745$; shCTRL vs shCLU, $P=0.1963$). The data are presented as mean \pm SEM. ** $p<0.01$, *** $p<0.001$, ns for not significant. Statistical comparisons between groups were performed using unpaired two-tailed Student's t tests. Source data are available online for this figure.

- For Figure 4 and S2A, it would be helpful to show statistics on the individual comparisons. It is unclear if shCTRL is significantly different from shCLU and PKP3 only, or also from shPKP3 + Clu.

Response: Thank you for your suggestion. As suggested, we have provided the statistics of Figure 4A and Figure EV5A in our revised manuscript.

Figure 4. ...MDA-MB-468 tumor-bearing mice with PKP3 knockdown, CLU knockdown, or PKP3 knockdown combined with ectopic overexpression of CLU (n=5 per group): (A) Body weight (shCTRL vs shPKP3, $P=0.0029$; shCTRL vs shCLU, $P=0.0016$; shCTRL vs shPKP3+CLU, $P=0.0412$);

Figure EV5. ...MDA-MB-231 tumor-bearing mice with PKP3 knockdown, CLU knockdown, or PKP3 knockdown combined with ectopic overexpression of CLU (n=5 per group): (A) Body weight (shCTRL vs shPKP3, $P=0.0015$; shCTRL vs shCLU, $P=0.0012$; shCTRL vs shPKP3+CLU, $P=0.0260$);

- For Figure 6, what is "anti-PKP3" and "anti-CLU"? The paper says these are lentiviruses. Are they shRNA, blocking antibodies, or something else? Please clarify in the text and Methods.

Response: Thanks for the suggestion. They were shRNA. We will clarify these in the text and methods in our revised manuscript.

Dear Dr Liu, dear Dr Chen,

Thank you for submitting your revised manuscript (EMBOJ-2025-120556R) to The EMBO Journal, as well for your patience with our feedback. Your amended study was sent back to the referees for their scientific reassessment, and we have received reports from all of them, which I enclose below. As you will see, the reviewers state that the work has been substantially enhanced by the revisions and they are now broadly in favour of publication, pending minor amendments.

Thus, we are pleased to inform you that your manuscript has been accepted in principle for publication in The EMBO Journal.

Please carefully consider the remaining minor points raised by referee #2 by adding complementary information on the human patients contributing to the study to the manuscript text and revising data statistics display and annotation in the figures legends.

Also, we now need you to take care of a number of issues related to formatting and data presentation as detailed below, which should be addressed at re-submission.

Please submit a revised version of the manuscript using the link enclosed below, addressing the advisor's comments.

As you might have seen on our web page, every paper at the EMBO Journal now includes a 'Synopsis', displayed on the html and freely accessible to all readers. The synopsis includes a 'model' figure as well as 2-5 one-short-sentence bullet points that summarize the article. I would appreciate if you could provide this figure and the bullet points.

Thank you again for giving us the chance to consider your manuscript for The EMBO Journal, I look forward to hearing from you and receiving your final revised version of the manuscript.

Best regards,

Daniel Klimmeck

>> Please limit the keywords for your study to maximally five.

>> Author Contributions: Remove the author contributions information from the manuscript text. Note that CRediT has replaced the traditional author contributions section as of now because it offers a systematic machine-readable author contributions format that allows for more effective research assessment. and use the free text boxes beneath each contributing author's name to add specific details on the author's contribution.

More information is available in our guide to authors.
<https://www.embopress.org/page/journal/14602075/authorguide>

>> Correct the order of the manuscript sections as follows: Abstract / Keywords / Introduction / Results / Discussion / Methods / Data Availability / Acknowledgements / Disclosure and Competing Interests Statement / References / Main Figure Legends / Tables / Expanded View Figure Legends. The Data Availability section should be after the Methods.

>> Rename the 'Materials and Methods' section to 'Methods'.

>> Remove the dataset legends from the manuscript text.

>> Funding: please enter the following funding information into our online system: 'Jiangsu Provincial Medical Innovation Center (Grant No. CXZX202224)'.

>> References: adjust the reference format to EMBO Journal format, 10 authors et al. .

>> Data availability section: please remove the referee token for the NCBI dataset and make sure that the data are made publicly accessible. Add hyperlinks to the datasets.

>> Please revisit the source data for the study provided on the imaging data.

>> Avoid textual redundancy in the introduction, results and discussion sections with your earlier 2021 study (PMID 33643800).

>> Ethics: please add information on the human participants and informed consent given in the manuscript text.

>> Author checklist: Ethics section: add information on the human participants and informed consent given.

>> Consider additional changes and comments from our production team as indicated below:

- DAS

1. Please note that the specific URL for PRJNA1287256 dataset is not provided in the data availability statement.

- Figure legends:

1. Please note that the exact p values are not provided in the legends of figures 1E, H; 2J; 6H

2. Please indicate the statistical test used for data analysis in the legends of figures 1A, 2A

3. Please note that the box plots need to be defined in terms of minima, maxima, centre, bounds of box and whiskers, and percentile in the legends of figures 1E, H; 2J, 4B, E; 6E, H; EV5 B

4. Please note that information related to n is missing in the legends of figures 1A, E, H; 3C, D, I; 4A, B, E, F, G, H, I; 5C, D, F, I, J, K, L; 6D, E, H, I; EV1, EV2 A, C, E; EV3 A, B, E, F, G, H, I, J; EV4 I, J, U; EV5 A, B, D, E, F, G, H, J.

5. Please note that the scale bar needs to be defined for figure 6A

6. Please note that scale bar and its definition are missing for figure 5A.

7. Please note that the white arrow heads are not defined in the legend of figure 3H. This needs to be rectified.

Please use the link below to submit your revision:

Referee #1:

The authors have significantly improved the manuscript and addressed all points. I recommend the manuscript for publication now.

Referee #2:

The authors have done substantial additional work to address criticisms and questions raised during the first review round. Overall, I find the manuscript to be substantially improved.

There is new human data presented comparing cachectic to noncachectic breast cancer samples, which is an important addition. An important missing piece here, however, is a description of how they defined cachexia and the clinical characteristics of the patients included in the analysis. My original review suggested an analysis based on degree of weight loss (e.g. regression analysis or some other multivariate approach). While this is not a necessary analysis, at the very least there needs to be additional information about the participants, how the different groups were defined (with emphasis on what distinguishes cachectic from non-cachectic) and how other key clinical characteristics compare, as would be standard for any cross-sectional study.

The authors have included additional experimental details in the figure legends, including 'n' for in vivo experiments (although for some of the supplemental figures such as metabolic cage data) it required going to the supplemental data file to discern n. For many of the in vivo experiments, they have an n=5, which is low for these types of metabolic analyses. In addition, by my read of the metabolic cage supplemental files, it appears as though they used n=3, which is extremely low for a metabolic cage experiment. I would suggest if this manuscript moves forward that there be a requirement for greater clarity in the figures, which continue to not show the data points and to over utilize dynamite plunger plots with SEM, obscuring the variance in the data--i.e. it would be better if the reader does not have to look at the accompanying raw data files to understand the strength and rigor of the data.

Referee #3:

This manuscript received a highly critical review, but I believe the Authors have risen to the challenge and adequately addressed all Reviewer concerns. I had asked the Authors to provide more mechanistic detail on the connections between Clusterin, Per3, and Circadian Rhythms, and the Authors did this with additional studies with 14-3-3 and other circadian targets. Reviewer 2 had the most challenging review, but I think the Authors have sufficiently addressed it by adding additional information on different fat depots, and providing additional experiments on the mechanistic and metabolic consequences of fat loss. The Authors also provided appropriate revisions in response to all Minor Comments. I believe this work is of a sufficient technological advance over prior work that it warrants publication in EMBO Journal.

Referee #1:

The authors have significantly improved the manuscript and addressed all points. I recommend the manuscript for publication now.

We are grateful to the reviewer for everything.

Referee #2:

The authors have done substantial additional work to address criticisms and questions raised during the first review round. Overall, I find the manuscript to be substantially improved.

There is new human data presented comparing cachectic to noncachectic breast cancer samples, which is an important addition. An important missing piece here, however, is a description of how they defined cachexia and the clinical characteristics of the patients included in the analysis. My original review suggested an analysis based on degree of weight loss (e.g. regression analysis or some other multivariate approach). While this is not a necessary analysis, at the very least there needs to be additional information about the participants, how the different groups were defined (with emphasis on what distinguishes cachectic from non-cachectic) and how other key clinical characteristics compare, as would be standard for any cross-sectional study.

The authors have included additional experimental details in the figure legends, including 'n' for in vivo experiments (although for some of the supplemental figures such as metabolic cage data) it required going to the supplemental data file to discern n. For many of the in vivo experiments, they have an n=5, which is low for these types of metabolic analyses. In addition, by my read of the metabolic cage supplemental files, it appears as though they used n=3, which is extremely low for a metabolic cage experiment. I would suggest if this manuscript moves forward that there be a requirement for greater clarity in the figures, which continue to not show the data points and to over utilize dynamite plunger plots with SEM, obscuring the variance in the data--i.e. it would be

better if the reader does not have to look at the accompanying raw data files to understand the strength and rigor of the data.

We sincerely thank the reviewer for the thorough evaluation and valuable comments. Tumor, blood samples and WAT were collected from 20 female TNBC patients with cachexia with body mass index (<18.5) and 21 female TNBC patients without cachexia with body mass index (18.5-24). Healthy blood samples were obtained from 22 healthy female persons with body mass index (18.5-24) and from 21 healthy female person with body mass index (<18.5). The clinical sample collection and experiment were carried out in accordance with guidelines and protocols approved by the Ethics and Scientific Committees of First Affiliated Hospital of Soochow University or Affiliated Cancer Hospital of Nanjing Medical University. This information were provided in the Methods section. As reviewer 2 suggestion, we have provided the “n” in the figure legend to help to improve the strength and rigor of the data. We thank the Reviewer for everything.

Referee #3:

This manuscript received a highly critical review, but I believe the Authors have risen to the challenge and adequately addressed all Reviewer concerns. I had asked the Authors to provide more mechanistic detail on the connections between Clusterin, Per3, and Circadian Rhythms, and the Authors did this with additional studies with 14-3-3 and other circadian targets. Reviewer 2 had the most challenging review, but I think the Authors have sufficiently addressed it by adding additional information on different fat depots, and providing additional experiments on the mechanistic and metabolic consequences of fat loss. The Authors also provided appropriate revisions in response to all Minor Comments. I believe this work is of a sufficient technological advance over prior work that it warrants publication in EMBO Journal.

We thank the Reviewer for everything.

Dear Dr Liu, dear Dr Chen,

Thank you for submitting the revised version of your manuscript. I have now evaluated your amended manuscript and concluded that the remaining minor concerns have been sufficiently addressed.

I am thus pleased to inform you that your manuscript has been accepted for publication in the EMBO Journal.

Kind regards,

Daniel Klimmeck

Daniel Klimmeck, PhD
Senior Editor
The EMBO Journal
EMBO
Postfach 1022-40
Meyerhofstrasse 1
D-69117 Heidelberg
contact@embojournal.org

Please note that it is The EMBO Journal policy for the transcript of the editorial process (containing referee reports and your response letters) to be published as an online supplement to each paper. If you should prefer removal of any referee-only figures included in the point-by-point response(s), e.g. because they may still be used for future publication or because they have been reproduced from published work by others, please do let us know immediately via response email.

More information is available here: https://www.embopress.org/transparent-process#Review_Process